# Multi-Objective Design and Optimization of Hardware-Friendly Grid-Based Sparse MIMO Arrays

**DOI:** 10.3390/s24216810

**Published:** 2024-10-23

**Authors:** Suleyman Gokhun Tanyer, Paul Dent, Murtaza Ali, Curtis Davis, Senthilkumar Rajagopal, Peter F. Driessen

**Affiliations:** 1MulticoreWare Inc., #228, 4010 Moorpark Ave., San Jose, CA 95117, USA; gokhun.tanyer@gmail.com; 2Department of Electrical & Computer Engineering, University of Victoria, P.O. Box 1700 STN CSC, Victoria, BC V8W 2Y2, Canada; 3Innovtec Inc., 1312 Dunsterville Ave., Victoria, BC V8Z 2X1, Canada; 4Uhnder Inc., 3409 Executive Center Drive, Suite 205, Austin, TX 78731, USA; paul.dent@att.net (P.D.); murtaza@uhnder.com (M.A.); curtis@multicorewareinc.com (C.D.); 5Royal Enfield, 296, Rajiv Gandhi Road, Sholinganallur, Chennai 600119, India; senthil.tpk@gmail.com

**Keywords:** grid-based sparse MIMO arrays, array design and optimization, mitigation of mutual coupling, adaptive desirability function, grating lobe-free arrays, sidelobe reduction, machine learning

## Abstract

A comprehensive design framework is proposed for optimizing sparse MIMO (multiple-input, multiple-output) arrays to enhance multi-target detection. The framework emphasizes efficient utilization of antenna resources, including strategies for minimizing inter-element mutual coupling and exploring alternative grid-based sparse array (GBSA) configurations by efficiently separating interacting elements. Alternative strategies are explored to enhance angular beamforming metrics, including beamwidth (BW), peak-to-sidelobe ratio (PSLR), and grating lobe limited field of view. Additionally, a set of performance metrics is introduced to evaluate virtual aperture effectiveness and beamwidth loss factors. The framework explores optimization strategies for the partial sharing of antenna elements, specifically tailored for multi-mode radar applications, utilizing the desirability function to enhance performance across various operational modes. A novel machine learning initialization approach is introduced for rapid convergence. Key observations include the potential for peak-to-sidelobe ratio (PSLR) reduction in dense arrays and insights into GBSA feasibility and performance compared to uniform arrays. The study validates the efficacy of the proposed framework through simulated and measured results. The study emphasizes the importance of effective sparse array processing in multi-target scenarios and highlights the advantages of the proposed design framework. The proposed design framework for grid-spaced sparse arrays stands out for its superior efficiency and applicability in processing hardware compared to both uniform and non-uniform arrays.

## 1. Introduction

Sensor arrays are integral components in spatio-temporal signal processing across diverse fields, including electromagnetic, acoustics, ultrasonic, and seismic processing systems. Their utility spans fundamental technologies such as radar, sonar, navigation, wireless communications, electronic surveillance, and radio astronomy [1,2,3,4]. Among these, MIMO radar leverages multiple antennas to transmit diverse waveforms, allowing for the maximization of power in specific areas of interest. The investigation of antenna pattern design for MIMO radar has garnered significant attention in recent years [5,6,7,8,9]. Prevalent in various signal processing and communication applications are sparse signal samples in time, frequency, or space domains [10,11,12].

While conventional FFT-based techniques offer a foundation for array design, recent years have seen the emergence of novel methods that surpass them in terms of both complexity and optimality [13,14,15]. Sparse antenna arrays, in particular, have gained traction for their ability to offer improved angular resolution through their thinned configurations, which benefit both phased array radar and MIMO radar. In contrast to the well-known repetitive sidelobe patterns inherent in uniform linear arrays (ULAs) and uniform rectangular arrays (URAs), sparse arrays intentionally distribute spacing unevenly to mitigate grating lobes effectively. However, the thinning of the array introduces an increase in the overall sidelobe level [1,5]. In each respective field, similar tools, algorithms, and reconstruction methods have been developed for sparse signal processing. These include random sampling of bandlimited signals, compressed sensing (CS) [16], channel estimation in Orthogonal Frequency Division Multiplexing (OFDM) [15,17], and calibration for the mutual coupling between antenna elements [18,19,20,21,22,23].

A key challenge in the research and application of sparse arrays lies in balancing the trade-offs between mitigating grating lobes and managing elevated sidelobe levels. Additionally, mutual coupling between antenna elements remains a persistent issue, which, if unaddressed, can severely impact performance. In this work, without loss of generality, our focus is on automotive radars, which have undergone a significant transformation in recent years. Evolving from niche sensors, they have become standard even in middle-class cars. Radar is frequently recognized as the most suitable sensor for automated braking and pedestrian safety functionality, as required by Euro-NCAP ratings. The industry is benefiting from the adoption of new radar-on-chip technology, integrating both analog and digital circuitry on the same substrate, resulting in cost-effective products [24,25,26,27].

Despite advancements in sparse array design, there remains a gap in optimizing these arrays for practical, hardware-friendly implementations, particularly in automotive radar systems. This study addresses these challenges by introducing a novel design framework for multi-objective optimization of grid-based sparse MIMO arrays (GBSAs). Key innovations include an efficient initialization method for rapid convergence and the use of adaptive desirability functions to balance performance metrics. This comprehensive framework not only enhances array performance but also ensures feasibility for real-world applications, offering significant advancements by tackling issues like mutual coupling and efficient resource utilization. By addressing these challenges, the study benefits both academic research and practical radar system development, distinguishing itself as a significant contribution to the field of automotive radar applications.

The remainder of this paper is organized as follows: In Section 2, we categorize different types of antennas, providing a thorough overview of their classifications. This serves as the groundwork for understanding the various configurations discussed later in the paper. Section 3 introduces sparse MIMO radar systems, with a focus on their unique characteristics and advantages. The section elaborates on the construction of virtual arrays and the operational principles behind multiple-input multiple-output (MIMO) radar. This lays the groundwork for sparse array optimization techniques.

In Section 4, we derive a classical Fourier transform-based beamforming formulation for sparse arrays. This section addresses the calibration of imperfect elements and accounts for inter-element mutual coupling, which is crucial to ensuring the accuracy of the beamforming process. Steering vectors for targets are also defined, along with the beamforming matrix for both transmitter and receiver elements. Section 5 introduces a multi-objective design and optimization framework for sparse MIMO arrays. Key performance parameters are identified and analyzed. As these parameters often conflict with one another, simultaneous consideration of all responses is required [28]. To address this, we utilize the concept of desirability, first introduced by Harrington (1965), as a key optimization tool [29,30]. The desirability function (DF) maps each independent variable onto a [0, 1] scale, reducing the multivariate optimization problem into a simpler univariate problem through the desirability index (DI) [31]. An efficient initialization method and a dynamic, data-driven approach for real-time optimization are also proposed, which eliminates the need for preset hyperparameters. This novel adaptive method incorporates data-driven parameter tuning and continuous learning based on real-time feedback, offering a basic form of dynamic optimization [14]. Furthermore, the joint optimization of multi-mode sparse arrays with complementary transmitter and receiver apertures is discussed, and a combination is explored here for the first time.

The effectiveness of grid-based sparse arrays is demonstrated through both simulated and experimental results in Section 6. Four distinct configurations are analyzed, confirming the benefits of the proposed optimization strategies. Finally, Section 7 concludes the paper by summarizing the advantages of grid-based sparse arrays and highlighting potential avenues for future research. Additional detailed mathematical formulations supporting this work are provided in Appendix A.

## 2. Classification of Antenna Arrays

Antenna arrays are classified based on inter-element spacing and design characteristics, as shown in Figure 1. An antenna array is considered uniformly spaced when the distance between elements is consistent in each orthogonal direction. In contrast, the array is classified as non-uniform if the spacing between elements is irregular or varies.

Uniformly spaced arrays can be represented in one dimension as a uniform linear array (ULA) and two dimensions as a uniform rectangular array (URA). Slanted URAs, where the axes are non-orthogonal, can be advantageous when the width and height of antenna elements differ significantly, as the slanted configuration allows the elements to be positioned more closely.

Thinned arrays are non-uniformly spaced arrays where the inter-element spacing values are quantized based on a minimum inter-element spacing that defines reference grid points. These arrays can be categorized into two types: dense and sparse arrays. In dense arrays, the average spacing between elements is less than one wavelength, while in sparse arrays, the average spacing exceeds one wavelength, as noted in [10]. The spacing of antenna elements can either be random or determined using a low discrepancy (LD) algorithm. In this context, ‘discrepancy’ refers to the deviation between a uniform element grid and randomized positions, as explained in [11] and further discussed in Section 5.2.*i*. While this concept applies to non-uniform arrays, we focus here on creating discrepancies in uniformly spaced arrays by selectively thinning the grid. The approach discussed in Section 3C is designed to facilitate the practical implementation of angle beamforming for sparse arrays. The antenna arrays analyzed in this paper are grid-based sparse arrays (GBSAs), and Figure 1 illustrates how GBSAs fit within the broader classification of antenna arrays.

## 3. Sparse MIMO Radar Systems

In a multiple-input multiple-output (MIMO) radar system, each transmitter (TX) signal is distinguished from every other TX signal by utilizing appropriate modulation differences, such as different digital code sequences. Each receiver (RX) correlates with each TX signal, generating several correlated outputs equal to the product of the number of RXs and the number of TXs. These correlated outputs can be interpreted as having been generated by several virtual receivers (VRXs) [4,6,7,9,32]. Common notations used throughout the paper are listed in Table 1.

MIMO processing enables the physical length to double in both directions, resulting in a four-fold increase in aperture area. The construction of virtual sparse arrays with different aperture sizes is demonstrated to be highly beneficial in achieving high-precision broadband direction finding [33]. Sparse arrays offer techniques for extremely large-scale (XL) arrays, providing potential solutions for overcoming severe path loss in millimeter-wave (mm-Wave) and Terahertz (THz) bands [34].

The antenna aperture and the desired field of view defined in the spherical coordinate system are illustrated in Figure 2. This standard coordinate system is described in [35,36,37,38].

### 3.1. Creation of Virtual Arrays

The MIMO processing performed at the range correlator enables the received signals due to differently encoded transmitters to be separated, leading to the creation of virtual receivers for each transmitter-receiver pair. The positions for these virtual arrays can be calculated analogously to the convolution operation, taking into consideration the relative positions of transmitter (TX) and receiver (RX) elements. Specifically, virtual arrays are generated by replicating transmitter arrays through each receiver, accounting for their relative positions. A visual comparison in Figure 3a,b illustrates that each receiver generates a replica of the transmitter array, perfectly aligned to form a virtual uniform linear array. In general, the number of virtual receivers equals the product of the number of transmitters and receivers. However, this count may decrease in instances where some of the virtual receivers (VRXs) overlap.

The relative positions between the TX and the RX elements illustrated by (a, b) and shown in Figure 3a are observed to create a physical shift for the virtual array shown in Figure 3b. This shift introduces a common complex phase factor in radiation equations which is often ignored.

### 3.2. Transmitting Grid-Based Sparse Antennas

The antenna aperture is commonly shared by transmitting and receiving elements, and the physical positions for uniform linear and rectangular arrays are well established. In contrast, sparse array elements, in general, can be positioned anywhere inside a given physical aperture. However, the elements of grid-based sparse arrays (GBSAs), as proposed in this context, are constrained to specific points defined by a reference grid space, as illustrated in Figure 3.

The radiation equation for a grid-based sparse array can be formulated by selectively disabling some of the transmitting elements in (A8) and introducing the uniform antenna element radiation pattern
(1)$$s_{s}\left(u,v\right)=\sum_{m=0}^{M-1}\sum_{n=0}^{N-1}r_{tx,s}\left(m,n\right)f_{tx}(u,v)e^{-j2\pi \left(md_{\lambda ,y}u+nd_{\lambda ,z}v\right)},$$
where $\left(u,v\right)=(sin\phi \sin\theta ,cos\theta )$, $\left(\phi ,\;\theta \right)$ represents the direction of the field point in the spherical coordinates, $r_{tx,s}\left(m,n\right)$ is the amplitude of the excitation current at the transmitter element, $d_{\lambda ,y}=d_{y}/\lambda$, and $d_{\lambda ,z}=d_{z}/\lambda$ are the inter-element spacing values, $f_{tx}(u,v)$ is the radiation pattern for the transmitting elements, and where $r_{tx,s}$ is nonzero only for a selection of (m,n) couples for active sparse elements (transmitters) located on the reference grid space.

Without loss of generality, (1) can be simplified by assuming isotropic antenna patterns and setting half-wavelength spacings $d_{\lambda ,y}=d_{\lambda ,z}=1/2,$
(2)$$s_{s}\left(u,v\right)=\sum_{m=0}^{M-1}\sum_{n=0}^{N-1}r_{tx,s}\left(m,n\right)e^{-j\pi \left(mu+nv\right)}.$$

### 3.3. Receiving Grid-Based Sparse MIMO Antenna Arrays

Similar to (2), (A11) can also be simplified to obtain the received signal at the virtual receiver elements
(3)$$r_{rx,s}\left(p\right)=\sum_{t=1}^{T}\sigma_{c,t}\;e^{j\pi \left(mu_{t}+nv_{t}\right)}$$
for $p=1,\;2,\;\ldots P_{s}<MN\;$ are the active VRXs created by available TX and RXs located on the grid points (m,n).

## 4. Angle Beamforming

Recalling the properties of the Fourier transform and its inverse discussed in Appendix A and using (2), and (3), the received signal pattern for a fully populated uniform array at the direction (u,v) can be calculated by
(4)$$g\left(u,v\right)=br$$
where the received signal at all available VRXs forms the received signal (column) vector ***r*** of length P, and b is the steering (column) vector evaluated at $\left(u,v\right)$. One can oversample $\left(u,v\right)$ uniformly for the received signal pattern
(5)$$u_{m}=\frac{2m^{\prime }}{Mq_{\phi }}-1,\;\mathrm{for}\;0\leq m^{\prime }<Mq_{\phi },$$
(6)$$v_{n}=\frac{2n^{\prime }}{Nq_{\theta }}-1,\;\mathrm{for}\;0\leq n^{\prime }<Nq_{\theta }.$$

The corresponding non-uniform samples in the angular domain can be calculated using the inverse-projection equations given below (see Figure 4)
(7)$$\phi_{m^{\prime },n^{\prime }}=\mathrm{asin}\left(\frac{u}{\sin\left(\theta_{m^{\prime }}\right)}\right)=\mathrm{asin}\left[\frac{\frac{2m^{\prime }}{Mq_{\phi }}-1}{\sqrt{1-{\left(\frac{2n^{\prime }}{Nq_{\theta }}-1\right)}^{2}}}\right],$$
(8)$$\theta_{n^{\prime }}=\mathrm{acos}v=\mathrm{acos}\left(\frac{2n^{\prime }}{Nq_{\theta }}-1\right)$$
where the above equation is valid only for real angles satisfying $u^{2}+v^{2}\leq 1.$ Here, $q_{\phi }$ and $q_{\theta }$ are positive integer oversampling factors for the *ϕ*- and *θ*-axes, respectively (Figure 4).

Now let us sort those angular samples into a single (row) vector, and evaluate the steering vector at each row, yielding the angle beamforming matrix ***B***. The received signal pattern can be calculated as
(9)g=Br
where g is the column vector for the received signal pattern, g is calculated in (4), r is defined in (3), and where ***B*** is the angle beamforming matrix with a size $\left[\left(M-1\right)\left(N-1\right)q_{\phi }q_{\theta }\right)]$ by P. For the hardware implementation of a multi-resolution ***B***, both oversampling ratios, $q_{\phi }$ and $q_{\theta }$, can be set to be multiples of 2 and the largest of those can be used to calculate a vector look-up table for the list of the orthogonal set of Fourier coefficients.

## 5. Multi-Objective Design and Optimization of Grid-Based Sparse Arrays

For a grid-based sparse array, both the physical and virtual array elements are thinned, and $r\left(p\right)$ is nonzero only for its $P_{s}$ elements where $P_{s}<P$ [14,38,39,40]. Deleting the zero, Equation (9) simplifies to
(10)$$g_{s}=B_{s}r_{s}$$
where both columns of $B_{s}$, and rows of $r_{s}$ are reduced to $P_{s}$, and the *thinning/sparsity ratio* is defined as $t=P_{s}/P\leq 1$, and where for a fully populated array, this thinning ratio is 1.

### 5.1. Optimizing Parameters

In practice, radar design engineers often face the challenge of satisfying multiple conflicting objectives for the same desired antenna array. The main optimization constraints include the following:A usable field of view (uFOV): The maximum grating lobe-free angular extent around the broadside, beyond which lies a flipped replica of the interior pattern that carries no additional information about the target.Beamwidth (BW): The desired angular width of the main lobe of the antenna pattern.Total number of physical elements (NTX and NRX): The count of both transmitting (TX) and receiving (RX) elements in the antenna array.Peak-to-sidelobe ratio (PSLR): A measure of the maximum amplitude of the main lobe relative to the sidelobes.

In addition to those, here are practical considerations.

*Physical size limitations:* The TX and the RX antenna elements in practice have finite physical dimensions, namely their width and height, which impose constraints on the minimum inter-element spacing values. These spacing values limit the horizontal and vertical uFOV values, respectively. Densely packed ULAs and URAs are directly affected by this limitation, especially when any size dimension is larger than
λ/2.*Mutual coupling:* The TX and RX groups should often be physically separated to decrease inter-group mutual coupling [1]. Mutual coupling among the same types of elements is assumed to be calibrated digitally.*Antenna element sharing of different arrays:* In multi-functional radars, some of the TX and RX elements are often shared between different scan modes. The antenna array design and optimization for all scans need to be performed simultaneously. A practical approach involves forcing the physical elements for a simpler scan mode to be used in some other complicated antenna configuration, effectively utilizing the array aperture.*Hardware implementation constraints:* Antenna elements are fed by transmission lines or waveguide structures, usually implemented on a separate neighboring hardware board. The layouts of transmitted and received signals should also be fed from another layer. As a design choice, a central region can be preferred to keep all the transmission lines approximately equal in length. This central region needs to be defined as a forbidden zone for the array elements.

In this work, thinning of fully populated uniform MIMO antenna arrays is examined to form effective sparse arrays, with the focus on improving the usable field of view (uFOV), beamwidth (BW), and peak-to-sidelobe ratio (PSLR) using fewer physical antenna elements simultaneously. The detailed definitions for these parameters are provided in the following subsections.

i.Peak-to-sidelobe ratio (PSLR):

Considering a given target range and velocity, the peak-to-sidelobe ratio (PSLR) is an important metric. The maximum skin return for a target is given by
(11)$$g_{peak}=\left|g_{s}\left(\phi_{t},\theta_{t}\right)\right|=\underset{\phi ,\theta \;\mathrm{in}\;\mathrm{FOV}}{\max}\left|g_{s}\left(\phi ,\theta \right)\right|$$
where $\left(\phi_{t},\theta_{t}\right)$ is the direction of the brightest signal observation in the FOV. The maximum sidelobe level is given by
(12)$$g_{msll}=\left|g_{s}\left(\phi_{s},\theta_{s}\right)\right|=\underset{\phi ,\theta \;\mathrm{in}\;\mathrm{FOV}}{\max}\left|{\tilde{g}}_{s}\left(\phi ,\theta \right)\right|$$
where $\left(\phi_{s},\theta_{s}\right)$ is the direction of the largest sidelobe in the uFOV, and where ${\tilde{g}}_{s}$ is obtained by setting $g_{s}$ to zero inside its main lobe region. This sidelobe has the potential to create false targets, even when it is outside of the operational FOV (oFOV), as long as oFOV ≤ uFOV.

The peak-to-sidelobe ratio (PSLR) can be calculated by
(13)$$PSLR\left(\phi_{t},\theta_{t};\phi_{s},\theta_{s}\right)=20\;log_{10}\left(\frac{g_{peak}}{g_{msll}}\right).$$

Note that for uniform and isotropic elements, assuming mutual coupling to be shift-invariant, and ignoring interference between the skin returns of neighboring targets in the presence of multi-targets, the PSLR becomes independent of the target direction.
(14)$$PSLR=20\;log_{10}\left(\frac{g_{peak}}{g_{msl}}\right).$$

ii.Beamwidth for uniform arrays:

The first null, and the half-power beamwidth can be calculated as follows [41]
(15)$$BW_{fn}=\mathrm{asin}\left(\frac{1}{L_{\lambda }}\right),$$
(16)$$BW_{hp}=0.886\left(\frac{1}{L_{\lambda }}\right)$$
where N is the number of uniform linear array (ULA) elements, $BW_{hp}$ and $BW_{fn}$ are the half-power beamwidth (HPBW) and the first null beamwidth calculated in radians, and where $d_{\lambda }$ and $L_{\lambda }=\left(N-1\right)d_{\lambda }$ are inter-element spacing and aperture length in terms of wavelength, respectively. Equations (15) and (16) are valid for uniform arrays for their horizontal and elevation cross-sections, respectively.

iii.Sidelobes, Grating lobes, and Usable FOV for sparse arrays:

ULAs and URAs, by definition, have constant inter-element spacings, and at any target angle, the inter-element phase differences are also constant, ideally zero at the target angle, and all elements contribute constructively. However, for field angles retreating from the main lobe moving along one of the axes, these phase differences with respect to the array center begin to rotate for the furthest elements the fastest. The first null occurs when the two elements furthest from the array center are out of phase with respect to this center by approximately ±π, yielding the first minimum. These far elements are $d\left(N-1\right)/2$ m away from the array center with a phase difference of $\pm \pi d\_\lambda \;\left(N-1\right)$. The first null is observed approximately at $\pm sin^{-1}\left(1/d_{\lambda }\;\left(N-1\right)\right)$ radians, respectively. This approximation is accurate within 0.6, 0.4, 0.2, and 0.1 degrees for *N* = 15, 18, 25, and 35, respectively. Retreating further from the broadside angle, while the furthest elements’ phase rotations turn back to in phase, we observe the second to the last elements to be approximately out of phase creating the second minimum. In this narrow angular region bounded by those two minima, all except the last two elements are mostly in phase, causing the largest sidelobe. Sidelobes gradually decrease due to increasing incoherence between the elements until we reach halfway to the first grating lobe. For an odd-numbered ULA, all radiations cancel each other completely, leaving the only one in the center to yield the minimum sidelobe level of 1/*N*. This phenomenon occurs independent of the number of elements or their spacings, and this first sidelobe sets the PSLR approximately to −13.8 dB at ϕ≈10.8° for $sinc\left(u\right)$ where u=2/2N.

Sparse arrays find applications in diverse fields, including astronomy [42,43] and autonomous vehicle radars. A notable sparse array in New Mexico prompt a discussion on its description and implications for Discrete Fourier Transform (DFT) requirements [44]. Unlike uniform arrays, sparse arrays, including the one in New Mexico, lack positions that are accurately rounded to integer values. Consequently, hardware implementations demand precise calculations of complex numbers for each angle beam, emphasizing the need to reduce the required number of complex coefficients. This reduction is effectively achieved by employing a uniform grid space.

In non-uniform array designs, the challenge arises from working in an infinite space, making solutions elusive with certain types of random searches. However, the use of grid-based sparse arrays proves advantageous, enabling a reduction in the search space/domain to a manageable size. Unlike non-uniform arrays, which typically require discrete Fourier transform (DFT), grid-based sparse arrays (GBSAs) utilize Fast Fourier Transform (FFT) with a practical list of beamforming coefficients. GBSAs offer a more efficient approach, especially when considering the number of coefficients needed for a specific field of view.

Simple heuristics aid in understanding the interactions between uniform linear array (ULA) and uniform rectangular array (URA) elements. These heuristics also elucidate why sparse arrays exhibit much lower skin return near (and even inside) the expected main lobe. The irregularity of sparse element positions leads to more than just two elements being out of phase away from the main lobe, as illustrated in Figure 5. In-phase interactions between elements rapidly become incoherent, resulting in a much smaller beamwidth. Despite this, due to the conservation of radiated power, the sidelobes cannot be entirely suppressed but spread across the half-hemisphere. Alternatively, they can be optimized to be pushed outside of a given field of view (FOV). This optimization is particularly crucial when dealing with a narrow FOV.

Grating lobes manifest at angles when *all* array elements are perfectly in phase, constructively contributing to the received signal pattern equivalent to the main lobe. Large sidelobes, distinct from grating lobes, necessitate only a comparatively large complex sum of all contributions, allowing for some inter-element phase mismatching and even some destructive interferences between elements.

For a fully populated ULA, a target located at some $\phi_{t}$ ($\theta_{t}=0$) creates grating lobes at angles given by
(17)$$\phi_{gl}=\mathrm{asin}\gamma_{n},\;\mathrm{for}\;-1<\gamma_{n}\;\leq 1,\;\mathrm{and}\;\mathrm{for}\;n=0,\pm 1,\;\pm 2\;\ldots$$
where $\gamma_{n}=\left(n/d_{\lambda }+\sin\phi_{t}\right)$ and $\phi_{gl}\neq \phi_{t}$, which yields the usable FOV (uFOV)
(18)$$\phi_{uFOV}=\pm \mathrm{asin}\left(\frac{1}{2d_{\lambda }}\right),\;\mathrm{for}\;-1<\frac{1}{2d_{\lambda }}\leq 1,\;\mathrm{and}\;\mathrm{for}\;n=\pm 1,\;\pm 2\;\ldots$$

For $d_{\lambda }=\lambda /2$, and $\phi_{t}=\pi /2$, the first grating lobe occurs for n=−2 at $\phi_{gl}=-\pi /2$, allowing the usable angular extension with $\phi_{ufov}=\left\{-\pi /2,\pi /2\right\}$ (Figure 2). Similarly, for $d_{\lambda }=\lambda$, $\phi_{t}=\pi /6,$ the closest grating lobe occurs for n=−2 at $\phi_{gl}=-\pi /6$, and the usable angular region becomes $\phi_{ufov}=\left\{-\pi /6,\pi /6\right\}$ as suggested by (17) and (18). It is crucial to note that, in practice, the physical size of elements limits the minimum possible inter-element spacing values both horizontally and vertically. With an element width of two wavelengths, the azimuthal uFOV is approximately restricted to ±14.48°, as shown in Table 2.

For field angles along the azimuth, −π<ϕ≤π, as illustrated in Figure 2, and for $d_{\lambda }\leq \lambda /2$, the condition $-\pi <kdu=2\pi d_{\lambda }u\leq \pi$ holds where *u* and *v* are defined in (1). In this scenario, each of the observed samples, −1<u≤1, carries independent information, and no grating lobes are observed. Equivalently, this discussion is valid for θ≤π and $-\pi <kdv=2\pi d_{\lambda }v\leq \pi$ where each measurement at the field point $\left(\phi ,\theta \right)$ is unique. However, for larger inter-element spacings, $d_{y}=\mu /2$ for μ>1, the condition $\left|\;kdu\right|\leq \pi$ is no longer satisfied. This confines the uFOV to −1/μ<u<1/μ and phase-wrapped target images could be created outside of this region. Conversely, real targets outside the uFOV also create their corresponding ‘target images’ inside the uFOV, making it challenging to distinguish real targets based on radar detections. This becomes a significant issue if the overall antenna patterns for the virtual receivers are not sufficiently directive to protect the uFOV.

‘Target images’, or equivalently, the grating lobes for a specific real target, emerge at angles relative to its source target angle, as shown in Figure 6. For instance, with a single broadside target, $d_{\lambda }=3/2$, and μ=3, the replicas of the target-return patterns are created in regions −1<u<−1/3 and 1/3<u<1 outside of the uFOV, as depicted in Figure 6a, with corresponding azimuth angles shown Figure 6b. Notably, at least three target patterns exist in the half-sphere regardless of the real target angle Figure 6c,d.

Target separations remain constant in the $\left(u,v\right)$ domain for varying target angles. The dotted reference line indicates the locus of target angles, $\phi =\phi_{t0}$, where the first phase-wrapped target image appears on the opposite side of the uFOV edge at $\phi =-\phi_{t0}$. Thus, uFOV $=2\phi_{t0}$ defines the range $-\phi_{t0}<\phi <\phi_{t0}$. In general, ($N_{t}-1)$ target images are created, separated by π/μ radians, where $N_{t}$ is the largest integer satisfying $N_{t}\leq \mu$. Target image separations decrease for larger μ, increasing the total number of images, as illustrated in Figure 6c,d.

Table 2 presents a concise list of calculated spacing values and their corresponding uFOV angles, providing valuable insights for design engineers. For forward-looking (FLR) and long-range radar (LRR) applications with narrow operational FOVs (<30°), large inter-element spacings can accommodate the use of large antennas with widths, $w_{\lambda }\leq 2$. In mid-range radar (MRR) scenarios with a uFOV=140°, smaller antenna elements with $w_{\lambda }\leq 0.5321$ are preferable. Short-range radar (SRR) applications requiring a wider uFOV face challenges when using large antenna elements, particularly when mitigating inter-element mutual coupling is a concern.

### 5.2. Design and Optimization of Grid-Based Sparse Arrays

The preceding discussion illustrates that ULAs and URAs can often become unpractical when confronted with constraints such as beamwidth (BW), uFOV, and element size limitations. Additional degrees of freedom become necessary to address these challenges. Sparse arrays, though known for generating undesired grating lobes [45], are purposefully designed in this work to offer optimal solutions by leveraging their irregular element positions. Moreover, sparse arrays offer the advantage of achieving exceptionally high angular resolution by enabling the use of significantly larger apertures. In this section, a practical design procedure for grating a lobe-free sparse array with high angular resolution is proposed.

Examining Section 5.1, it becomes evident that for uniform linear (ULAs) and rectangular arrays (URAs), the improvement of the beamwidth (BW) for a desired uFOV BW can only be achieved by increasing the antenna length $\left(L_{\lambda }\right)$, aperture size $\left(A_{\lambda }\right)$, or the number of elements $\left(N\right)$. Therefore, with a fixed FOV and N, BW improvement is not possible with ULAs and URAs. This limitation arises since the required FOV and BW values determine the size of the fully populated aperture, which may exceed practical limits. Furthermore, additional constraints, such as the physical size of elements and the mutual coupling, add complexity to the design process. There is currently a need for an efficient approach to design TX/RX arrays utilizing thinning, allowing larger apertures with fewer elements, without compromising the initial constraints significantly.

Without loss of generality, let us assume that the maximum number of elements and the limitations dictate the available number of array elements. Operational requirements may necessitate a desired BW value for angular resolution. The only initial constraints are the operating frequency and the desired physical antenna aperture size, while all other parameters remain open for optimization. The type of radar scan, whether a one-dimensional horizontal or a supporting two-dimensional elevation scan, determines the array dimension. Hardware implementations, technology, and cost determine the element size requirements. Initial simulations for mutual couplings provide insights into the required minimum horizontal and vertical separation distances, $L_{mc,y}$ and $L_{mc,z}$, to minimize these effects, which are discussed in Section 5.2.*ii*. Physical and virtual element sharing from a previously designed array (or for a multi-mode radar) could enforce some positions as initial values for the optimization (See Algorithm 1).

**Algorithm 1:** Sparse array optimization***desired parameters***:• array dimension (1D/2D/3D) and let us assume 1D as an example.• available number of TX and RX elements, *N_tx_*, *N_rx_*, respectively,• uFOV, and BW both for azimuth and elevation,***constraints***:• physical dimensions of antenna elements, *w_tx_*, *h_tx_*, *w_rx_*, *h_rx_*• forbidden zones for mutual coupling, *y_mc_*, *z_mc_*• enforced element positions***initialize k*** = 0:• calculate the required virtual aperture length, 2*Y_λ,max_* using (27) and (28)• calculate the required minimum inter-element spacings, ∆*d_λ,min_* using (29) and (30)• determine reference grid space, *y_n_*• set positions to the enforced list of positions• (optional HIA): calculate a uniformly distributed set of ∆*d_λ_*
$d_{\lambda }^{n}$***iterate k until*** (***k*** < ***K_max_***) ***or*** (***PSLR_best_*** > ***PSLR_desired_***)• random shuffling and random perturbations of
$∆d_{\lambda ,y}^{n}$• add non-overlapping TX and RX positions satisfying constraints until all positions are calculated• calculate the received signal pattern, and calculate the PSLR,• update array positions and *PSLR_best_* if *PSLR_new_* < *PSLR_best_*
**
*end*
**
***repeat:*** outer loop is used if desired and hyperparameters are also optimized for a given range of values.

In this initial step, the desired BW and uFOV values determine the minimum aperture length/size and the minimum inter-element spacing, respectively. Notably, at this step, there is no constraint for the element sizes to be smaller than the spacing values. This results in the largest possible grid spacing for the reference fully populated virtual array using (16) and (18) [46,47]. Readers can refer to the literature on various approaches to constructing such virtual arrays [6,7]. The available number of TX and RX elements determines the targeted thinning ratio as discussed in (10).

Initial element positions are enforced within the same grid space, where these positions can either be randomly selected or calculated using a low-discrepancy method, as discussed in the next section. As an iterative process, a random or preferred search algorithm is performed to determine successful arrays based on PSLRs and BWs, ensuring that each candidate satisfies all constraints. The iteration terminates upon achieving the desired PSLR or reaching the maximum number of iterations. Additionally, the desired and constructed hyperparameters can be optimized using an outer loop. The complexity of the optimization algorithm is studied in Section 6.4. While there are various approaches for improved convergence, including evolutionary algorithms and gradient descent-type analytical approaches [14], these will be explored in detail in future studies. Here, we propose a heuristic approach to illustrate rapid convergence to a satisfactory ‘local optimum’ array.

i.Low-discrepancy (LD) Inter-element Spacings:

A highly effective heuristic initialization approach ensures an even distribution of inter-element spacing values across the entire range, distinguishing itself from uniform linear arrays (ULAs) and uniform rectangular arrays (URAs). Unlike the repetitive patterns in ULAs and URAs that lead to distinct constructive and destructive interactions among radiations, sparse arrays intentionally disperse spacing distributions to effectively suppress grating lobes. The use of an initial value with a uniform distribution promotes fast convergence, especially beneficial for smaller sparsity values where the search domain size grows exponentially. This heuristic initialization approach results in a ‘good’ local array, directing the search path toward better candidates. In many instances, the optimum result is found in the vicinity of this initialization after just a few iterations.

To achieve a low PSLR, grating lobes must be spread outside the operational FOV. This sets the grid sizes according to (19) and Table 1 [48]. Subsequently, element positions are optimized for a reduced PSLR and the maximum allowable BW value(s). When the frequency of a specific inter-element spacing value is significantly larger than others, it can lead to a large sidelobe with high in-phase contributions. To mitigate this, irregular spacing values (Algorithm 1) can be employed to distribute large side lobes and grating lobes across the half-hemisphere. However, this may cause an overall rise in sidelobe levels due to the conservation of total radiated power. Grating lobe mitigation can be achieved by enforcing the empirical cumulative distribution function (ECDF), $F_{D}\;\left(d_{n}\right)$, representing a smooth density and, ideally, a uniform density where the ECDF is a staircase function with uniformly increasing steps, $F_{D}\;\left(d\right)=n/N$ for $d_{n}\leq \;d<d_{n+1}$, and $1<n<N-1,\;F_{D}\;\left(d\right)=0$ for $d<d_{1}$ and $F_{D}\;\left(d\right)=1$ for $d_{N}\leq d$.

One method of initializing spacing values for low discrepancy is to force their ECDF to be linear, as proposed below.

A linear GBSA with N elements has $\left(N-1\right)$ inter-element spacing values, $d_{n}$, which can be set to be empirically uniform.
(19)$$d_{n}=d_{min}+\left(n-1\right)\Delta d\;\;\;\;\mathrm{for}\;n=1\ldots \left(N-1\right)$$
where $d_{min}$ is the minimum inter-element spacing, for N≥3
(20)$$\Delta d=\frac{X_{max}-X_{min}-\left(N-1\right)d_{min}}{\sum_{k=1}^{N-2}k}$$
and where *N* elements are located between $X_{max}$ and $X_{min}$. It is important to note that it needs careful consideration when Δd is not a multiple of the desired grid spacing. This is studied in Section 6.4.

Let us begin by setting the reference grid space to λ/2 for a grating lobe-free pattern, disregarding the element size constraint at this stage. For $L_{\lambda }=11$ and N=5, we calculate Δd=3λ/2, and spacings are linearly increasing as given by $2d_{n}/\lambda =\{1,\;4,\;7,\;10$}, yielding element positions $2x_{n}/\lambda =\left\{0,\;1,\;5,\;12,\;22\right\}$ for $X_{min}=0$. This results in a linear ECDF, $F_{D}\left(d_{n}\right)=\left(1+d_{n}/\lambda \right)/6$, ensuring that all positions fall within the reference grid space.

Next, using the same reference grid space, for an element size of 2λ to avoid any overlap, we need $d_{min}=2\lambda$. Then, we calculate Δd=λ/2, and spacings linearly increase as $2d_{n}/\lambda =\{4,\;5,\;6,\;7$}, yielding positions $2x_{n}/\lambda =\left\{0,\;4,\;9,\;15,\;22\right\}$. This also results in a linear ECDF, $F_{D}\left(d_{n}\right)=\left(-3+2d_{n}/\lambda \right)/4$. No grating lobes are expected since the ECDF is smooth, and the positions are located on a uniform grid size of λ/2. It is worth noting that a random shuffling of the spacing order will change the element positions but not the ECDF. Each alternative configuration will yield to a different signal-return pattern, requiring search iterations for the optimum spacing ordering for the best PSLR. Perturbing the spacing values during search iterations is another strategy.

Similar to the second example, let us now set $L_{\lambda }=10$. Recalculating Δd=λ/3, spacings are given by $3d_{n}/\lambda =\{6,\;7,\;8,\;9$} with element positions, $3x_{n}/\lambda =\left\{0,\;6,\;13,\;21,\;30\right\}$, which does not coincide with the reference grid points. For a general sparse array, there is no constraint on the element positions as proposed in Table 2. However, a ‘uniform’ sparse array (GBSA), as defined in this paper, requires all its element positions to be in some uniform grid space. In this third example, we can change the reference grid size to λ/3 or move the elements to some neighboring grid points without causing any overlapping. Both approaches will provide the first group of initial array candidates for the optimization search. Element positions can be further perturbed to move around the neighboring grids, finding a better ‘local optimum’ PSLR value as described in Table 2. The ECDF for the optimum spacings is expected to be a smooth function with minimal repetitive spacing values to spread the grating lobes into sidelobes. The ECDF is further examined in Section 5.3.

ii.Sparse arrays with minimized mutual coupling:

The sparse array design procedure described above allows for flexible positioning of both TX and RX elements within a shared aperture, which is particularly advantageous for designing smaller BWs. However, it is crucial to address potential challenges related to high mutual coupling between TX and RX groups. In cases where mutual coupling can significantly impact the system’s beamforming performance, it might be necessary to physically separate the two groups, allocating different sections of the aperture to each. This separation helps mitigate the negative effects of mutual coupling, ensuring optimal performance.

To study this scenario, the beamforming Equation (4) is modified to account for mutual coupling contributions
(21)$$g\left(u,v\right)=\int_{S}C\left(x_{\lambda },y_{\lambda };u,v\right)\;r\left(x_{\lambda },y_{\lambda }\right)\exp\left[j2\pi \left(x_{\lambda }u+y_{\lambda }v\right)\right]dx_{\lambda }\;dy_{\lambda },$$
and physical aperture sparsing can be performed as in (10)
(22)$$g_{s}=B_{s}C_{s}\;r_{s}$$
where $C_{s}$ is the square mutual coupling matrix of size $P_{s}$. The optimization of sparse arrays can be further constrained by introducing a ‘forbidden zone’ that mandates a physical separation between TX and RX elements. This additional constraint simplifies (22) by assuming that the mutual couplings become negligible when distances between TX and RX elements exceed specific thresholds, denoted as $y_{mc}$ and $z_{mc}$ for the horizontal and vertical dimensions, respectively. The ‘forbidden zone’ enforces a spatial segregation, mitigating mutual coupling effects and contributing to the overall optimization of the sparse array.

iii.Efficient design of virtual arrays:

The compensation for imperfections arising from mismatched antenna elements can be achieved through separate digital calibrations for the TX and RX antenna groups, as discussed in [18]. However, mitigating the effects of transmitter-to-receiver coupling is a more intricate process, dependent not only on target variables but also on the field angle. In addressing this complex mutual coupling, a strategy involves physically separating the transmitter (TX) and receiver (RX) groups and implementing forbidden zones on the physical aperture. This approach minimizes mutual coupling issues but demands careful consideration during the sparse array design process. Further attention is required to avoid poor placement of transmitter and receiver elements, which could lead to aperture loss and potentially result in degradation of both PSLR and BW. Thus, a thorough understanding and management of these factors are crucial for successful sparse array optimization.

Alternative approaches to defining forbidden zones for minimizing mutual coupling between the TX and RX elements are shown in Figure 7. In this illustration, a physical aperture of 40λ×30λ and a uniform grid spacing of λ/2 are considered. Minimum inter-element distances between element groups are assumed to be 15λ and 10λ along the azimuth and elevation, respectively. In the case of uniform arrays, the virtual aperture is created by doubling both the horizontal and vertical physical lengths, leading to a virtual aperture size four times that of the corresponding physical dimensions. All the array structures depicted in Figure 7 adopt a 40λ × 30λ configuration and virtual apertures are anticipated to fall within an 80λ × 60λ maximum possible area. The effect of the forbidden zones is observed to create lost virtual apertures, as expected. Accounting for the physical element sizes to prevent overlapping renders certain positions unusable, leading to further aperture loss and beamwidth spreading, necessitating careful consideration.

Let us define the performance metrics for virtual aperture and BW efficiency.

*Virtual Aperture Efficiency* ($\alpha_{ap}$):

(23)$$\alpha_{ap}=(\beta_{v}A_{phy})\;/A_{vrx}$$
where $A_{phy}$ and $A_{vrx}$ represent the physical and virtual aperture sizes, respectively. The virtual aperture gain, $\beta_{v}$, takes values 2 and 4 for the 1D and 2D cases, respectively. The aperture loss factor, $\alpha_{ap}\leq 1$, assesses the efficiency of the aperture.

*Beamwidth Spreading Factor* ($\alpha_{bw}$):

(24)$$\alpha_{bw}=BW_{observed,hp}/BW_{theory,hp}$$
where $BW_{observed,hp}$ and $BW_{theory,hp}$ denote the observed and theoretical half-power beamwidths, respectively. The beamwidth spreading factor ($\alpha_{bw}\leq 1$) is calculated for both azimuth and elevation in Table 3 for each configuration in Figure 7. These metrics provide a quantitative evaluation of the performance of the sparse array, considering aperture loss and beamwidth spreading considerations.

The fully populated reference array is most efficient in creating the largest possible virtual aperture, providing a unity aperture loss factor. In theoretical considerations, standard ULAs and URAs utilize TX and RX elements that can be co-located on a shared aperture, and their physical sizes are often ignored, as shown in Table 3 (a, b). However, in practice, this is not possible due to the physical size of the antenna elements. For practical implementations, larger, more suitable options are highlighted in bold for clarity.

The aperture loss factor as defined in (23) represents the ratio of the utilized virtual area to the maximum possible virtual area and gives unity for the reference URA. The available physical apertures are often divided into TX and RX sub-apertures to minimize their mutual couplings and to avoid overlapping. Different design rules for defining sub-apertures are illustrated in Table 3 (c–n), providing a pool of positions for the algorithm given in Algorithm 1. The efficiency of the improved four-corners case is further enhanced in Table 3 (m, n) compared to the four corners in Table 3 (g, h). Note that $\alpha_{ap}\leq 1$ decreases as the separation distance between the TX and RX groups increases.

The half-power beamwidth (HPBW), as defined in (15) and (16), is a measure of the angular resolution that improves with a larger aperture. Once again, the reference URA provides the smallest value. The HPBW spreading factors, $\alpha_{\phi }\geq 1$, and $\alpha_{\theta }\geq 1$, are defined as the ratio between the observed HPBW and the minimum possible value given by the reference URA. The improved four-corners approach yields the best overall results for the aperture loss and the beamwidth loss factors compared to the given alternative approaches listed in Table 3. As part of future work, other alternative design approaches for achieving better performance metrics will be studied.

The thinning of arrays is widely recognized for its capacity to achieve superior angular resolution, even though it is acknowledged to lead to higher peak-to-sidelobe ratio (PSLR) values [45]. This characteristic is also evident in sparse arrays, as discussed in Section 5.1*.iii*., particularly when the sparsity ratio is small. Nonetheless, it is important that PSLR experiences a 5 dB reduction when the sparsity ratio falls within the range of 60 to 90 percent. This observation suggests that PSLR reduction is possible for dense arrays, as indicated by the dip in PSLR in the lower left section of Figure 8.

### 5.3. Multi-Objective Optimization of Sparse Arrays Using the Desirability Function

The desirability function (DF) is a powerful optimization technique, particularly useful when managing multiple conflicting objectives. It combines different objectives into a single composite desirability value, enabling the identification of optimal solutions that address all objectives simultaneously (see Section 5.4 for further details).

The DF assigns a score to each response based on how closely it matches the desired target. The simplest form of the DF is linear, assigning a score to each response based on its distance from the desired target. However, more complex forms can capture nonlinear relationships between the objectives, offering a more detailed evaluation of the optimization process [14].

While the algorithm proposed in Section 5.1 focuses on improving a single parameter, as illustrated for the PSLR in Algorithm 1, real-world antenna design often involves multiple conflicting requirements. Improving one parameter may negatively affect others, such as achieving a low PSLR at the expense of a wider beam width. The desirability function is valuable in these cases, as it enables the simultaneous optimization of multiple constraints [14]. Below, the one-sided DF is illustrated for three parameters: PSLR, and beamwidths along both the azimuth and elevation.
(25)$$D_{O}={\left(\overset{N}{\underset{i=1}{\prod }}d_{i}\right)}^{1/\gamma_{0\;}},$$
(26)$$d_{v}^{+}=\left\{\begin{array}{cc} 0 & v\leq v_{min}\\ {\left[\frac{v-v_{min}}{v_{desired}-v_{min}}\right]}^{\gamma_{v}} & v_{min}\leq v\leq \;v_{desired}\\ 1 & v_{desired}\leq v \end{array}\right.,$$
(27)$$d_{v}^{-}=1-d_{v}^{+}D_{O}={\left(\overset{N}{\underset{i=1}{\prod }}d_{i}\right)}^{1/\gamma_{0\;}},$$
where $D_{O}$ is the overall desirability function (desirability index) for *N* observing parameters, $\gamma_{v}$ and $\gamma_{0\;}=\sum \gamma_{i}$ are individual and overall geometrical weight coefficients for the desirability functions, $d_{v}$ and $D_{0}$, respectively, and for the parameter v where $d_{v}^{+}$, and $d_{v}^{-}$ promote larger and smaller values, respectively, and where v, $v_{min}=v_{undesired}$, and $v_{desired}$ are the observed, minimum (undesired) acceptable value, and the desired value for some variable *v*.

For the case studied here, smaller PSLR and BW values are preferred. Consequently, $d_{PSLR}^{-}$, $d_{BW,\phi }^{-}$, and $d_{BW,\theta }^{-}$ are the selected smaller-the-better (STB) desirability functions for the PSLR, azimuth BW, and elevation BW, respectively. Thus, the overall desirability function becomes
(28)$$D_{O}={\left(d_{PSLR}^{-}\;d_{BW,\phi }^{-}\;d_{BW,\theta }^{-}\right)}^{1/\left(\gamma_{PSLR}+\gamma_{BW,\phi }+\gamma_{BW,\theta }\right)}.$$

It is evident that $0\leq D_{O}\leq 1$ holds since the inequalities $0\leq d_{i}\leq 1$ hold for each $d_{i},\;\left(i=1,\;2,\;\ldots N\right)$. One may assign equal weights to all three variables by setting $\gamma_{PSLR}=\gamma_{BW,\phi }=\gamma_{BW,\theta }=1$. Alternatively, for a different emphasis, one can prioritize PSLR over beamwidth along the azimuth and the elevation, with $\gamma_{PSLR}=2$, $\gamma_{BW,\phi }=1$, and $\gamma_{BW,\theta }=0.5$. Similar to (25)–(28), the LTB-DF can be constructed using the sigmoid function as illustrated in Figure 9a where $d_{v}^{+}=S\left(x\right)$, $S\left(x\right)=1/\left(1+e^{-x}\right)$, $x=10\frac{v-v_{m}}{v_{e}}$, $v_{e}=v_{max}-v_{min}$ and $v_{m}=(v_{max}+v_{min})/2$.

Equation (25) provides a valuable variable, with the desirability function (DF), for multi-objective optimization, serving as a metric to be monitored throughout the optimization process to identify the ‘best’ antenna array. The procedure proposed in Algorithm 1 can be adapted for any number of objectives simply by substituting $D_{best}$ in place of $PSLR_{best}$.

### 5.4. Adaptive Desirability Function for Learning of Hyperparameters

The concept of desirability, as defined in (25)–(27), and first introduced by Harrington (1965), serves as a key optimization tool [29,30]. The desirability function (DF) maps each independent variable onto a [0, 1] range, where multiple responses, potentially with different units or ranges, are converted into a single, unitless desirability value. These values are then combined to form an overall desirability function, index, or score [31,49,50]. This framework simplifies the multivariate optimization problem into a more manageable univariate problem, enhancing its practical application. Typically, (25)–(27) can be implemented as single or two-sided functions [51]. The desirability function is often selected as ‘lower the better’ (LTB) or ‘larger the better’ (LTB), where the goal is to maximize the score toward 1 or 0. In some cases, the overall desirability index is calculated as the sum of individual DFs [52].

In optimization, defining an accurate desirability function (DF) requires prior knowledge of the trade-off parameter ranges and the hyperparameters $v_{undesired}$ and $v_{desired}$ for each variable. The accuracy of this information significantly impacts optimization performance, and convergence may be hindered if the target bounds are incorrectly defined. Initialization of the DF plays a crucial role in the convergence rate, and designing an effective DF can be a time-consuming process.

In the initial step, all DFs should be constructed as defined, with careful attention given to setting the hyperparameter values for each variable. The dynamic range, calculated by $\left|v_{undesired}-v_{desired}\right|$ should be broad enough to make the changes visible at each epoch, allowing the measurement of convergence. However, to achieve optimal results, the dynamic range should gradually narrow toward the desired target. This creates a conflict in hyperparameter settings, often requiring manual tuning to balance the visibility of changes and convergence efficiency, which increases the overall number of epochs. To overcome this, it is crucial to use the adaptive desirability functions (ADFs) that dynamically adjust hyperparameters, rather than relying on fixed values. The algorithm iteratively adjusts the variable ranges based on incoming data, essentially ‘learning’ the optimal range during each epoch of the optimization process. This continuous adaptation is guided by the relationship between the variables and the responses, allowing the algorithm to refine its behaviors as it encounters more data.

This adaptive process can be viewed as a basic form of machine learning, specifically, online learning. Here, the algorithm does not rely on a fixed model but rather updates its parameters dynamically in response to new data. By adjusting the variable ranges and desirability parameters at each iteration, the adaptive desirability function learns to optimize the responses more effectively over time. While simpler than more advanced machine learning algorithms, this method shares key characteristics, such as data-driven parameter tuning and continuous learning based on real-time feedback, making it a basic form of machine learning algorithm [14].

The practical approach proposed in [14] for determining an adaptive desirability function (ADF) can be applied here. In this context, a machine learning approach is proposed, utilizing the adaptive desirability function (ADF) given below to ensure consistent progress during optimization without requiring prior knowledge. The one-sided larger-the-better (LTB) DF, DF, $d_{v}^{+}\left(k+1\right)$, for the $\left(k+1\right)$’th iteration is given by
(29)$$d_{v}^{+}\left(k+1\right)=S\left(10\frac{v\left(k+1\right)-v_{m}\left(k\right)}{v_{e}\left(k\right)}\right),$$
where $v_{e}$, $v_{m}$, and *S* are defined earlier, and the desired parameter range is continuously updated using the most recent observations,
(30)$$v_{min}\left(k\right)=\min\left\{v_{min}\left(j\right)\right\},\;\;\mathrm{for}\;\;k-10\leq j\leq k-1,\;$$
(31)$$v_{desired}\left(k\right)=\max\left\{v_{desired}\left(j\right)\right\},\;\;\mathrm{for}\;\;k-10\leq j\leq k-1.$$

### 5.5. Disadvantages of Sparse Arrays

The optimum sparse array design of Section 5.2, as illustrated in Figure 5, suggests uniformly distributed inter-element spacings, effectively attenuating and spreading the grating lobes into sidelobes. Consequently, sparse arrays can provide solutions with no grating lobes. However, this comes at the cost of spreading the grating lobe power across the uFOV, leading to increased sidelobe levels and reduced PSLR. This effect becomes more pronounced, especially for wider uFOV and smaller thinning ratios, as depicted in Figure 9. In general, sparse arrays often require effective specialized processing techniques to mitigate the reduction in PSLR, particularly in multi-target scenarios. Various practical approaches addressing this challenge are readily available [24,53,54].

## 6. Results

### 6.1. Fully Populated Uniform Arrays

Grating lobes, uFOV, and BW are examined in Figure 10. The received signal patterns illustrate the number of VRXs, inter-element spacings, aperture lengths, uFOV, and BW for the cases (a) 121, λ/2, 60λ, 180°, 0.3875, (b) 121, λ, 120λ, 90°, 0.1937, and (c) 61, λ, 60λ, 90°, 0.4039, respectively. The inverse proportionality between BW and $L_{\lambda }$, as suggested by (15) and (16), holds. Additionally, for $d_{\lambda }\geq 1/2$, grating lobe angles and uFOV angles demonstrate an inverse proportionality, as indicated by (17) and (18). It is noteworthy that PSLR remains approximately constant for these cases. To simultaneously improve uFOV and BW, an increase in the number of elements is necessary, with only one degree of freedom.

The target signal-return pattern is observed to shift without any distortion in the $\left(u,v\right)$ plane, as shown in Figure 11a,b, where the signal is hard-limited from below −70 dB for enhanced visual clarity. This underscores the primary reason for the preference for the $\left(u,v\right)$ plane over the $\left(\phi ,\theta \right)$, particularly when evaluating sidelobes arising from distinct targets in a multi-target scenario.

### 6.2. Grid-Based Sparse Arrays (GBSA) with Large Antenna Elements

Here are some numerical results illustrating GBSAs with no grating lobes, optimized with or without mutual coupling forbidden zones. For the optimization, the A version’s 12 RXs and the B version’s 8 RX positions are enforced as the initial condition. The physical aperture size is limited to 35λ×35λ, and element sizes are 5λ×2λ for the so-called Ankara–1 array, and their versions A and B introduced in the next subsection.

i.GBSA with no forbidden zones: Ankara–1 A and B arrays

Ankara–1 A and B arrays represent the first two examples of GBSA with no forbidden zones. Versions A and B have 16 and 8 RXs, respectively. Version A is enforced to use B’s 8 RXs and 12 TXs, and its remaining 8 RXs (the first 8) are optimized separately.

Sparse Ankara A and B GBSAs are configured on a reference grid with dimensions λ/2×λ, employing physical element sizes of 2λ×5λ, ensuring freedom from grating lobes exclusively along the horizontal that do not have forbidden zones as illustrated in Figure 12a,b. A total of 12 TX and 16 RX elements share a common usable aperture of 30λ×30λ, with non-overlapping elements of size 5λ×2λ, where the usable aperture refers to the available region for the element centers. The optimization outlined in Algorithm 1 is performed in two stages. Initially, it is applied to Version B for 12 TX and 8 RX positions. Subsequently, a second run is conducted to calculate the remaining 8 RXs for Version A, using the same TX positions.

The received signal pattern for a single target in the direction $\left(\phi_{t},\theta_{t}\right)=\left(30{}^{\circ},\;30{}^{\circ}\right)$ is depicted in Figure 12c,d, while the broadside for the azimuth and elevation cross-sections is presented in Figure 12e–h. Notably, no observed horizontal grating lobe is present, given that $d_{x}=\lambda /2$. However, a target at an elevation angle of 30°, located at the elevation uFOV edge, generates its image at −30°, consistent with the expectations outlined in Table 2. PSLR is noted to be below −10 dB and −8 dB within the uFOV for Version A and B, respectively, in accordance with the optimization hyperparameter.

The maximum PSLR values for arrays A and B are 11.23 dB and 8.64 dB, respectively. The azimuth and elevation HPBW values are [1.1°, 1.1°] and [1.0°, 1.0°], respectively.

ii.Inter-element mutual coupling for sparse Ankara-1 array:

Inter-element mutual coupling within an antenna array is significantly influenced by the specific positions of both transmitters (TXs) and receivers (RXs). For elements of the same type, compensations are primarily achievable through laboratory measurements and signal calibration. However, coupling among those two groups strongly depends on the environment and the target angles. The simulated results for the Ankara array are shown in Table 4. The coupling values, denoted by K, range from −116.2 dB to −36.3 dB. The physical positions of the array elements are shown in Figure 12a, while coupling values are illustrated in Figure 13. Values smaller than −60 dB are ignored here due to the presence of background spurious imperfections at that level. The maximum and minimum coupling values, −36.3 dB and −116.2 dB, correspond to the coupling between the couples TX-7 $\left(22\lambda ,\;9\lambda \right):$ RX-11 $\left(28.5\lambda ,\;9\lambda \right)$ and TX-1 $\left(30\;\lambda ,\;30\;\lambda \right)$: RX-15$\left(24.5\lambda ,\;\lambda \right)$ with separation distances 6.5λ and 29.5λ, respectively. Notably, the minimum horizontal and vertical distances of 15λ and 10λ between the element centers emerge as preferable separation distances for minimizing mutual coupling. This constraint will be rigorously applied in the subsequent section.

iii.GBSA with forbidden zones: Ankara–2 A and B arrays

This section demonstrates the application of the improved four-corners approach proposed in Figure 7m,n to the design of Ankara–2 arrays. To minimize coupling effects, forbidden distances of $y_{mc}=10\lambda$ and $z_{mc}=15\lambda$ are enforced. Elements and physical sizes in versions A and B remain identical to Ankara–1 GBSAs, offering improved array aperture and BW spreading efficiency as defined in Equations (23) and (24).

The Ankara–2 GBSAs are designed on a reference grid with dimensions λ/2×λ/2, utilizing physical element sizes of 2λ×5λ. The design ensures grating lobe-free operation both horizontally and vertically, as illustrated in Figure 14c–h. The minimum PSLR values are observed to be 11.10 dB and 9.14 dB. The azimuth and elevation HPBW values are $\left[0.8{}^{\circ},\;1.5{}^{\circ}\right]$, $\left[1.0{}^{\circ},\;1.45{}^{\circ}\right]$ for A and B arrays, respectively.

The measured maximum PSLR outside of the main beam aligns with the simulated results, revealing reduced sidelobes attributed to the directivity of radar elements contributing to MIMO directivity. The measured BW is around 0.32 degrees, and a secondary wider main lobe within the (−10 dB, −6 dB) range extends to approximately 0.72 degrees. This two-tiered main lobe structure is clearly formed by the two separate groupings of the virtual array structure, as illustrated in Figure 14b.

The captured radar data for the received signal at the Doppler filter output for Ankara–2A is used to calculate the angle beamformer output presented in Figure 15 as a function of $\left[u,v\right]$. The angle values $\left(\phi ,\;\theta \right)$ corresponding to the $\left[u,v\right]$ values can be calculated using $\left(u,v\right)=\left(\sin\phi \sin\theta ,\cos\theta \right)$, where the values $\left[0,\;0.25,\;0.5,\;1.0\right]$ correspond to the angles $\left(0,\;14.48{}^{\circ},\;60{}^{\circ},\;90{}^{\circ}\right)$.

The novel GBSA design approach outlined in Section 4 offers effective array solutions with improved FOV and BW, especially for low numbers of elements. Ankara arrays enable the utilization of large element sizes, ensuring that the received signal patterns are grating lobe-free along any desired axis. The results illustrated in Table 5 demonstrate that achieving PSLR values greater than 11 dB and BW less than 0.5 degrees is possible with only $N_{tx}=12$, $N_{rx}=16$, and $N_{vrx}=192$. Additional measured performance values are provided in Table 3.

### 6.3. The Empirical Cumulative Distribution Functions (ECDFs) for the Inter-Element Spacings

The inter-element spacings for ULAs are constant as depicted in Figure 16a for both *N* = 16 and 64. The staircase function ECDF, as defined in Section 5.2.*i.*, simplifies to a single-step function due to the constant spacing value of $d_{\lambda }=1$, creating grating lobes. In the case of the grid-based sparse Ankara arrays with versions A and B, illustrated in Figure 16b, the minimum spacing for each vertical is also $d_{\lambda }=1$. This is because the closest possible distance between two elements is constrained by the width of λ. It is worth noting that antenna elements are freely positioned within the λ/2 grid space avoiding the grating lobes.

To ensure grating lobe-free operation, these spacings need to be λ/2 or smaller. However, when the physical size of elements and so the spacings exceed this value, the received signal pattern for ULAs becomes the sum of all image replicas for each real-valued grating lobe angle at $asin\left(n/2w\right)$ shifted along the azimuth. Similarly, for URAs, additional replica images will be added for each real-valued grating lobe at $asin\left(n/2h\right)$ shifted along the elevation, as discussed in (17). Additionally, avoiding large sidelobes is only possible if the sparse array has irregular spacing values. Irregular spacing occurs when the empirical cumulative distribution function (ECDF) lacks substantial increments in large values, and the step-increment values are relatively uniform, as shown in Figure 16b.

### 6.4. Hardware Efficiency of Grid-Based Sparse Arrays

For a general MIMO array, the beamforming equation is expressed as g=Br in (9) where g is the received signal pattern and r represents the Doppler filter output for each virtual array channel. For an array with 12 transmitters and 16 receivers, the number of virtual array elements is 192.

Assuming the transmitter and the receiver elements share a physical aperture of 30 λ×40 λ, as suggested in Figure 7m, the MIMO processing generates a virtual aperture of 60 λ×80 λ, with no aperture loss both for azimuth and elevation. The aperture loss factor $\alpha_{ap}$ is 1 when no mutual coupling zone is defined, ($y_{mp}=z_{mp}=0$, as provided in Table 3). The grid-based sparse array (GBSA) design allows for the utilization of the grid space of 120×160 with λ/2 sampling along both axes. Remarkably, only a few transmitters and receivers can generate a grid space of 120×160=19,200 virtual elements. For this specific array, the thinning (sparsity) ratio is 192/19,200×100=1%.

Next, let us assume that the desired output signal pattern is two-dimensional, with 512×256 samples (beams) for azimuth and elevation, respectively. In this case, g is a column vector with 512×256=131,072 row elements. The beamforming matrix ***B*** would have a size of 131,072×192, resulting in a total of 25,165,824 unique complex coefficients that need to be calculated or stored in the radar hardware for a general sparse array. This is particularly true for non-uniform (gridless) sparse arrays [39]. However, in the case of a grid-based sparse array (GBSA), the exponential coefficients become regular due to overlapping of the wrapped coefficients within the $\left\{0,\;2\pi \right\}$ range. As a result, the number of coefficients reduces to just 512. This reduction in storage by a factor of 49,152 enables the use of standard FFT hardware, offering a significant advantage in the hardware implementation of GBSAs.

The proposed design framework offers superior efficiency and practicality compared to both uniform and non-uniform arrays in hardware processing. Uniform arrays require a significantly larger number of physical elements and more complex feeding hardware to achieve the same beamwidth (BW) value. Similarly, non-uniform sparse arrays demand an excessive number of coefficients, making their implementation highly impractical. In contrast, the grid-based sparse array (GBSA) framework significantly reduces the number of required coefficients, making it far more feasible for real-world applications, especially in radar systems where efficient hardware implementation is essential.

### 6.5. Fast Convergent Sparse Array Optimization Using Structured Methods

The steps of the optimization, as outlined in Algorithm 1, begin with calculating the physical aperture based on a desired beamwidth (BW) value using (15) and (16). Once all design parameters, including the total number of antenna elements, are initialized, the first candidate array is generated. At each epoch, the performance metrics of the candidate array are calculated in sequence, starting with the received signal pattern, followed by the PSLR, and the BW, and finally the desirability index (DI) as the evaluation metric. Since the calculation of the received signal pattern is computationally costly, minimizing its frequency in the optimization process is crucial for reducing complexity. Equations (19) and (20) provide a strong initial candidate compared to a random array. This is achieved by enforcing the empirical cumulative distribution function (ECDF) to be uniform, helping to evenly distribute grating lobes within the usable field of view (uFOV).

It is important to note that (19) is ideally satisfied when the available physical aperture equals $\Delta d\left(N-1\right)$, where each inter-element spacing is exactly Δd. However, in many cases, the aperture requirements for a sparse array may differ, causing Δd to become a non-integer multiple of the desired grid spacing. The required quantization of Δd to be a multiple of the desired spacing creates a limited set of array candidates, which can be tested at each epoch. Additionally, introducing a random addition or subtraction of a single spacing to some inter-element spacing values further expands the set of candidate arrays. It is observed that the proposed optimization process efficiently reduces complexity by minimizing costly calculations through structured methods like uniform ECDF enforcement. Random spacing adjustments further expand the candidate set, improving performance while maintaining computational feasibility.

Further improvements could focus on enhancing the adaptability of the desirability functions (DFs) for multiple variables. In the current method, the DFs are updated independently at each epoch, resulting in a zig-zag convergence path. While this behaviour allows the optimization to explore a wider range of solutions, it may not always be efficient. A potential refinement could involve calculating the overall gradient at each epoch to guide the optimization toward a smoother convergence path. However, it is essential to first study whether the zig-zag pattern significantly delays convergence. If proven inefficient, implementing gradient-based adjustments could balance the progress across multiple variables, leading to faster and more stable optimization.

## 7. Conclusions

This study introduces a comprehensive design and optimization approach for sparse arrays, particularly emphasizing grid-based antenna element configurations. The proposed methodology addresses multi-objective optimization, considering higher PSLR, smaller BW, the challenge of the utilization of larger antenna elements causing grating lobes, and the critical issue of minimizing mutual couplings.

By defining grid-based sparse arrays, this study presents a systematic design framework that considers the trade-offs associated with peak sidelobe ratio (PSLR), beamwidth (BW), and mutual coupling. The proposed approach offers flexibility in positioning transmitter (TX) and receiver (RX) elements within a shared aperture, enabling the creation of virtual apertures with enhanced beamforming performance.

Multi-objective optimization is a key focus, allowing for the simultaneous consideration of conflicting objectives. The desirability function (DF) is introduced as an effective tool to balance multiple constraints, providing valuable insights for researchers and engineers working on sparse array applications. Furthermore, the study acknowledges challenges related to antenna elements exceeding sizes and creating grating lobes. The optimization process takes these challenges into account, ensuring that the resulting sparse arrays meet specific design requirements, particularly for smaller bandwidths.

A significant aspect of the proposed methodology is its capability to minimize mutual couplings between TX and RX groups. By introducing forbidden zones based on separation distances, the study encourages the physical separation of these groups, mitigating potential negative effects on beamforming performance.

In conclusion, this study offers a holistic approach to sparse array design and optimization, addressing various challenges associated with PSLR, BW, grating lobes, and mutual couplings. The presented framework provides a valuable foundation for advancing sparse array applications in diverse domains.

## 8. Patents

Related patents resulting from this work are reported in [46,47,53].

## Figures and Tables

**Figure 1 sensors-24-06810-f001:**
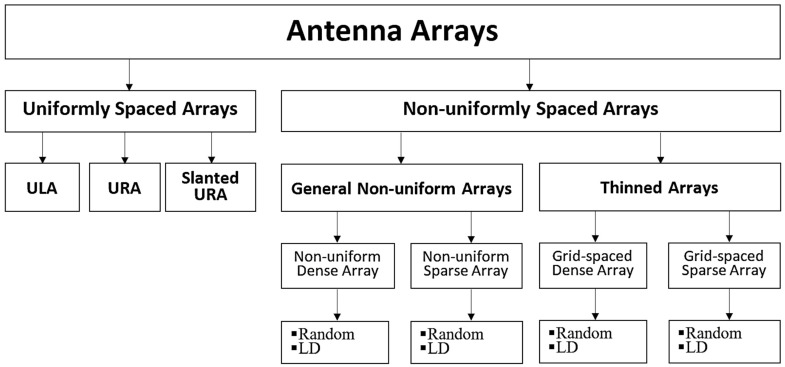
Classification of antenna arrays in terms of inter-element spacing values.

**Figure 2 sensors-24-06810-f002:**
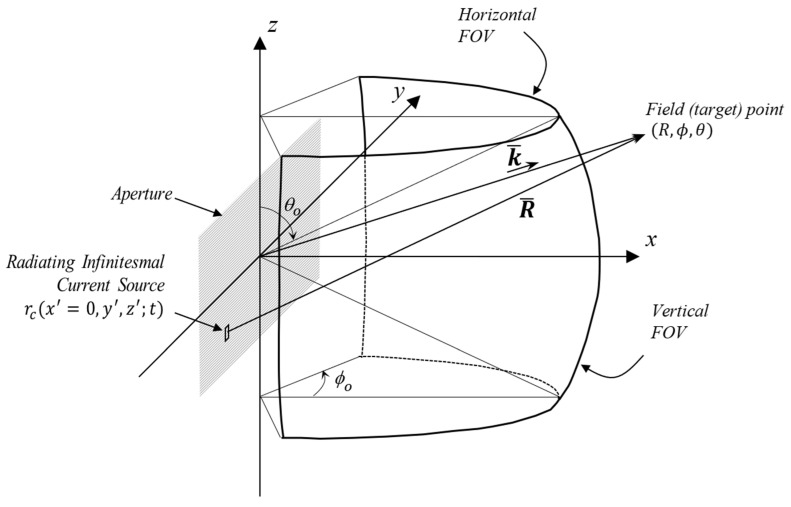
Problem geometry for a planar antenna aperture.

**Figure 3 sensors-24-06810-f003:**
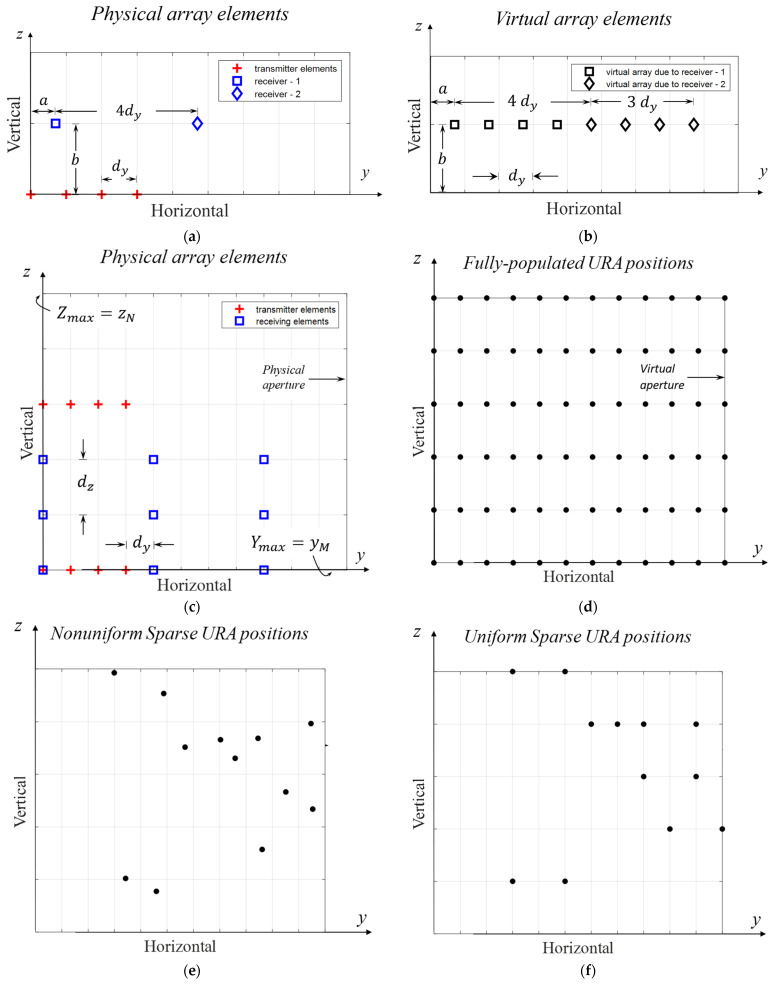
Examples for physical and virtual arrays. (**a**) The physical elements for 4 TX, 2 RX ULA antenna arrays, (**b**) 8 element virtual ULA created by the elements given in (**a**). (**c**) The physical elements for 8 TX, 9 RX URA antenna arrays, and (**d**) corresponding fully populated 2D URA virtual array of 72 elements located on the reference grid space. Two other examples for virtual arrays, (**e**) a non-uniform, and (**f**) grid-based virtual sparse arrays, respectively (physical locations are ignored).

**Figure 4 sensors-24-06810-f004:**
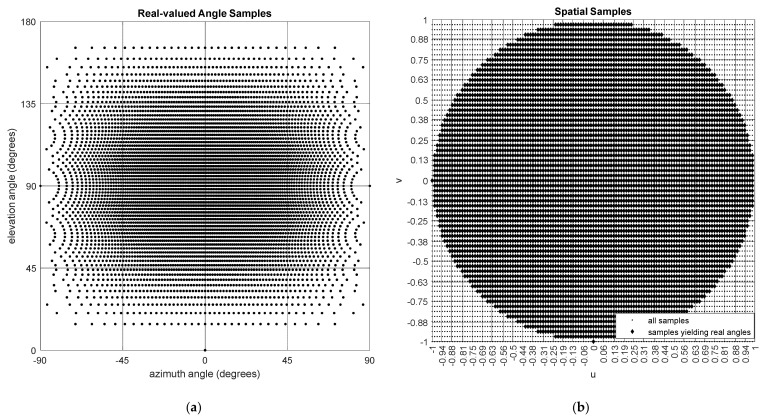
(**a**) Uniformly sampled spatial variables, and (**b**) the corresponding real angles for which (*u*^2^ + *v*^2^) ≤ 1 is satisfied.

**Figure 5 sensors-24-06810-f005:**
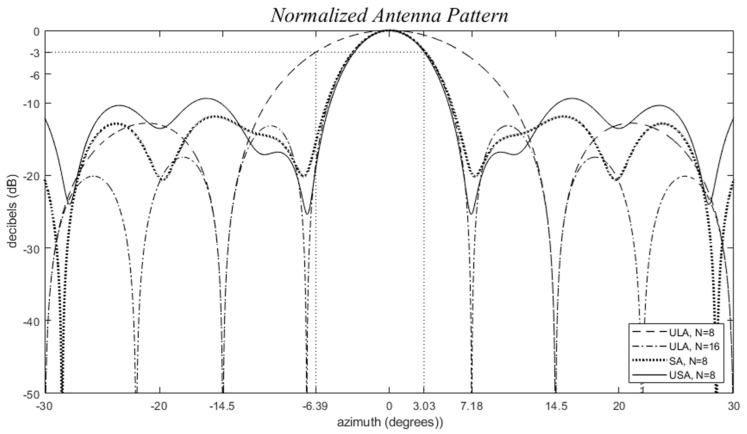
Uniform linear arrays (ULA), *d_λ_* = 0.5, (dashed) *N* = 8, (dashed dot) *N* = 16, and some examples to (dotted) a sparse array (SA) with *d_λ_* = 0.5, and *N* = 8 with nonuniform positions *d_λ_* = 0, 1.07, 2.52, 3.79, 4.48, 4.98, 5.86, 7.5, and (solid) a grid-based sparse array (GSA) with *d_λ_* = 0.5, and *N* = 8 using the positions of as reference for which are thinned to 0, 3, 5, 7, 8, 10, 14, and 15. This figure shows how the main lobe is spread when the array is thinned.

**Figure 6 sensors-24-06810-f006:**
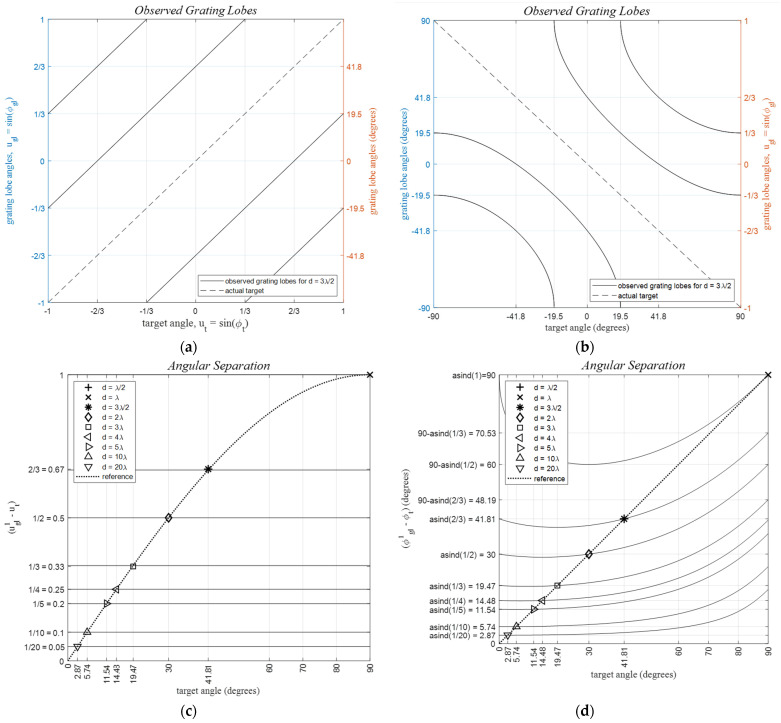
Grating lobe angles with respect to the target angle. (**a**,**b**) grating lobes with respect to the target angle for a ULA with *d_λ_* = 3*λ*/2, (**c**,**d**), angular separation between the first grating lobe angle, $u_{gl}^{1}$, and the target angle, *u_t_*, for various inter-element spacings where the dotted reference line shows the locus of target angles, $\phi =\phi_{t0}$, for which the first phase-wrapped target image appears on the opposite side of the uFOV at $\phi =-\phi_{t0}$. All values given in the vertical axes carry the angles (**left**) represented in the (*u*, *v*) plane, and (**right**) given in degrees, respectively. Note that for (**c**,**d**) there is no grating lobe for *d* = *λ*/2.

**Figure 7 sensors-24-06810-f007:**
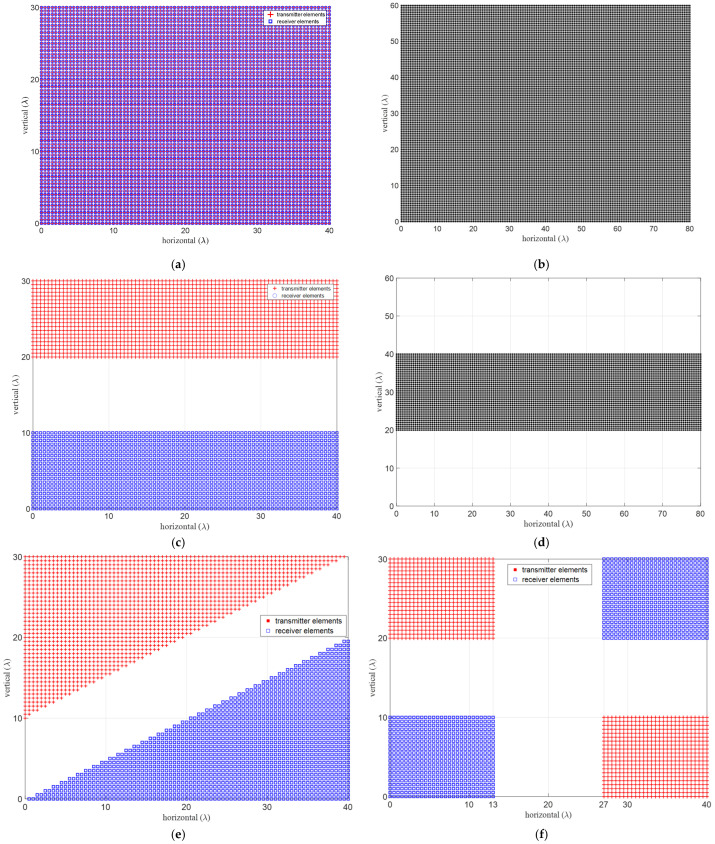
Example array structures with mutual coupling forbidden zones, (**left**) physical apertures for (red) TX and (blue) for RX where the opposite also yields the same VRXs, and (**right**) their virtual apertures, (**a**,**b**) fully-populated URA with no forbidden zones and co-located TX and RXs, (**c**,**d**) two-vertical, (**e**,**f**) two-diagonal, (**g**,**h**) four-corners and (**i**,**j**) thick-L shaped, (**k**,**l**) wrap-around, and (**m**,**n**) improved four corner structures, respectively. Forbidden distances, *y_mc_* and *z_mc_* are selected to be 15*λ* and 10*λ*, respectively.

**Figure 8 sensors-24-06810-f008:**
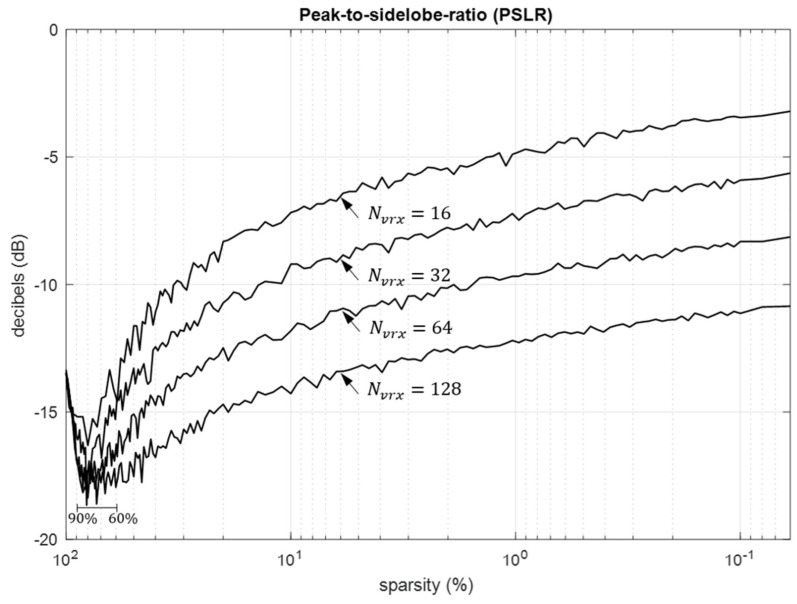
PSLR as a function of sparsity for ULAs with aperture lengths of 16*λ*, 32*λ*, 64*λ*, and 128*λ*.

**Figure 9 sensors-24-06810-f009:**
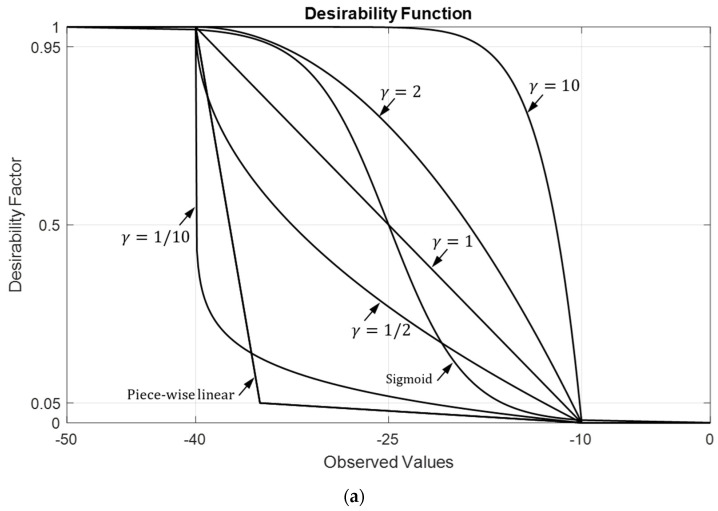
Single and two-variable LTB desirability functions, (**a**) linear, nonlinear (sigmoid function) and piece-wise linear functions with different weights, (**b**) $D_{PSLR}^{-}$ and $D_{BW}^{-}$ for $\gamma_{PSLR}=2$, and $\gamma_{BW}=0.5,$ and (**c**) $D_{0}=D_{PSLR}^{-}D_{BW}^{-}$. Desired regions are assumed to be $\left(-40\;\mathrm{dB}<PSLR<-10\;\mathrm{dB}\right)$, and $\left(0.25{}^{\circ}<BW<2{}^{\circ}\right)$.

**Figure 10 sensors-24-06810-f010:**
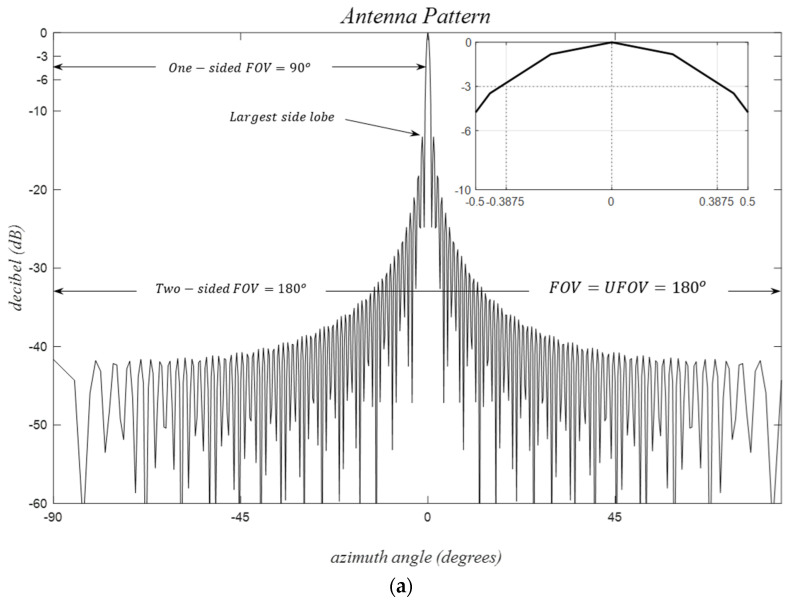
Received signal patterns for uniform linear MIMO arrays with different parameters; (**a**) *N_vrx_* = 121, *d* = *λ*/2, (**b**) *N_vrx_* = 61, *d* = *λ*/2, (**c**) *N_vrx_* = 61, *d* = *λ*.

**Figure 11 sensors-24-06810-f011:**
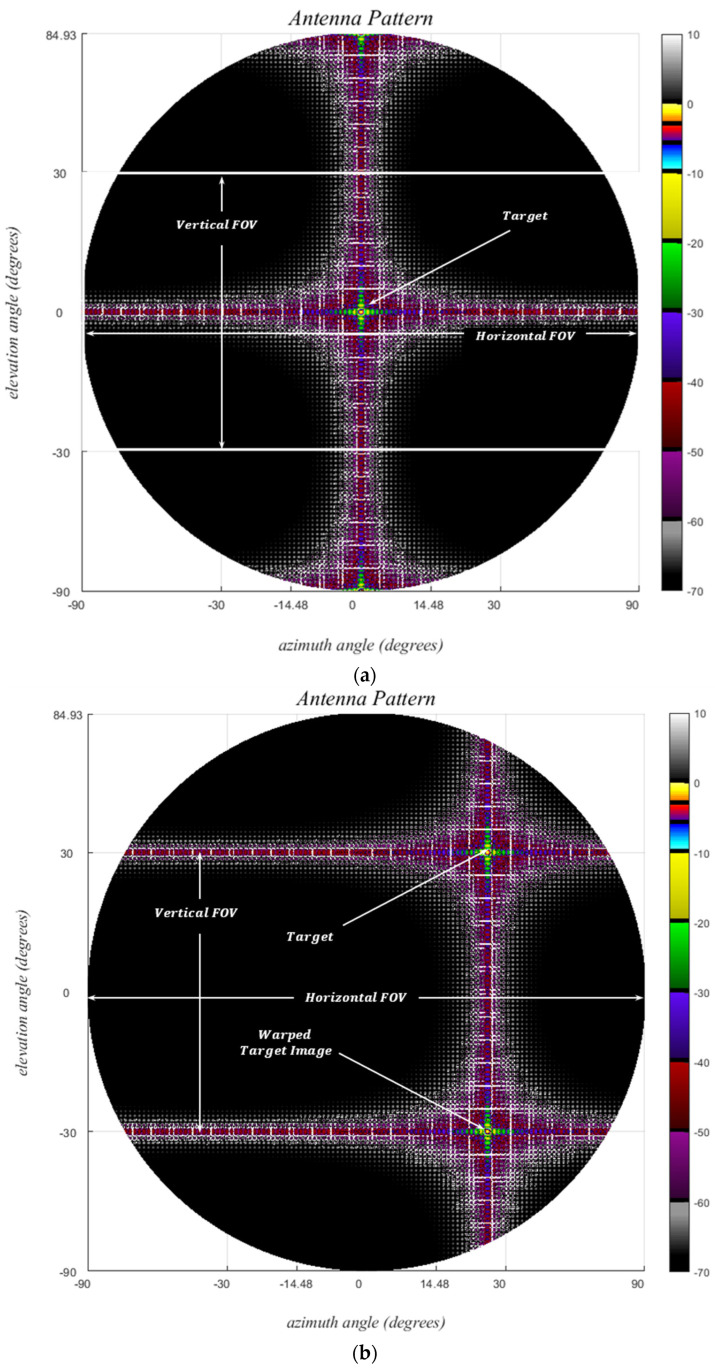
Received normalized signal patterns in decibels for a uniform rectangular MIMO array; (*M_vrx_*, *N_vrx_*) = (121, 61), *d* = *λ*/2, (**a**) (*ϕ_t_*, *θ_t_*) = (0, 0), (**b**) (*ϕ_t_*, *θ_t_*) = (0, 30°), (**c**) close up view of the beam width region.

**Figure 12 sensors-24-06810-f012:**
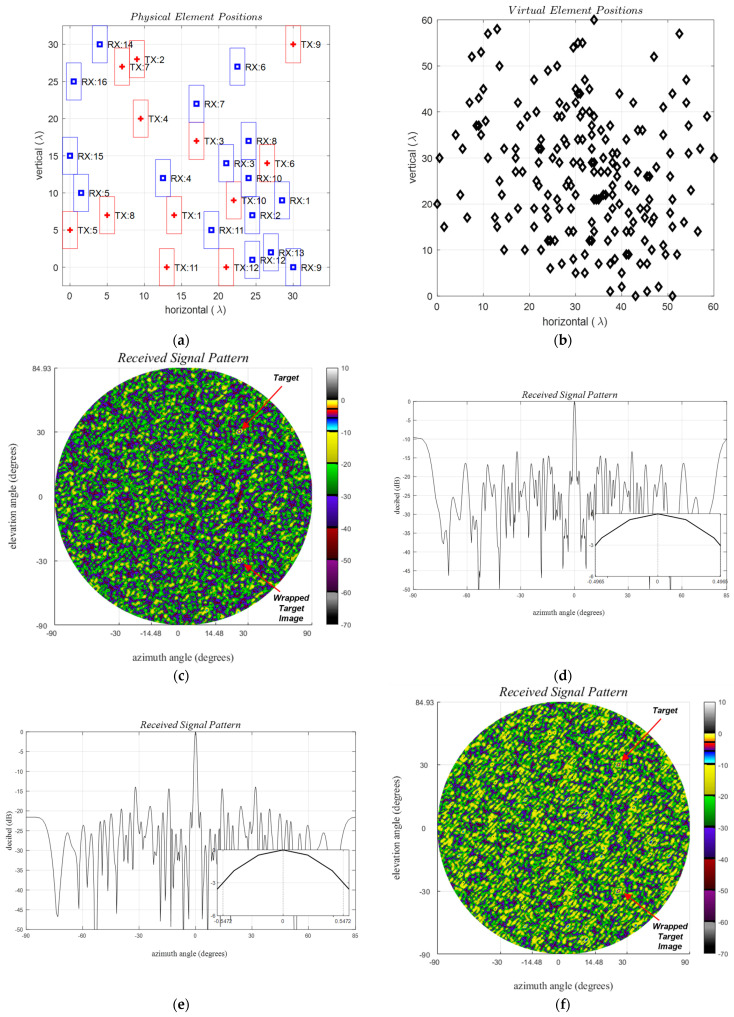
Grid-based sparse arrays, *d_y_* = *λ*/2, *d_z_* = *λ*, *w_tx_* = *w_rx_* = 2*λ*, *h_tx_* = *h_rx_* = 5*λ*, no forbidden zones are defined. (**left**) Ankara–1A, (**right**) Ankara–1B, (**a**,**b**) Physical element positions where Ankara–1A is utilizing all elements whereas for Ankara–1B the first 8 RXs are disabled, (**c**–**h**) 2D and 1D received signal patterns in the (*u*, *v*) planes with tick values converted to degrees. Single target angles, (*ϕ_t_*, *θ_t_*), are (30°, 30°), and (0°, 0°) for 2D and 1D plots, respectively.

**Figure 13 sensors-24-06810-f013:**
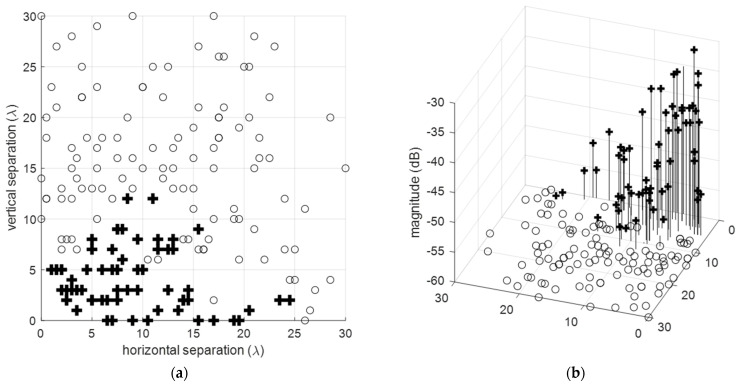
Inter-element mutual coupling between the transmitter and receiver groups for the sparse array Ankara-1. (**a**,**b**) calculated mutual coupling values, (+) *K* > −60 dB, (o) *K* < −60 dB.

**Figure 14 sensors-24-06810-f014:**
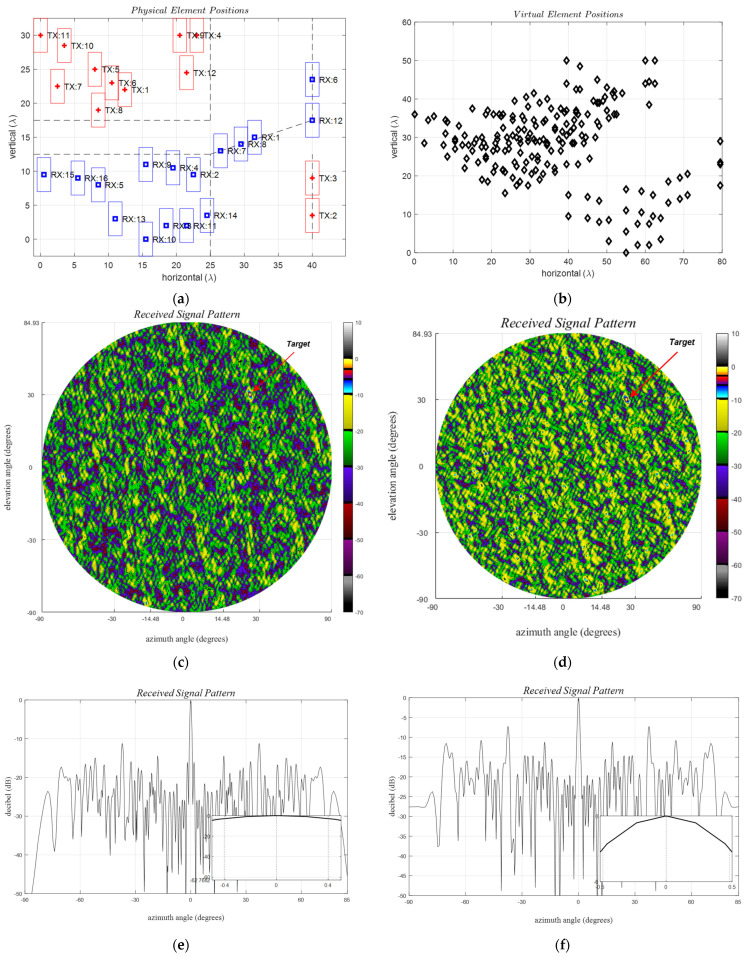
Mutual coupling constrained grid-based sparse arrays, *d_y_* = *d_z_* = *d* = *λ*/2, *w_tx_* = *w_rx_* = 2*λ*, *h_tx_* = *h_rx_* = 5*λ*, the forbidden distances, *y_mc_* = 10*λ*, and *z_mc_* = 15*λ*. (**left**) Ankara–2A, (**right**) Ankara–2B, (**a**,**b**) Physical element positions where Ankara–2A is utilizing all elements whereas for Ankara–2B the first 8 RXs are disabled, (**c**–**h**) optimized 2D and 1D received signal patterns in the (*u*, *v*) planes with tick values converted to degrees. Single target angles, (*ϕ_t_*, *θ_t_*), are (30°, 30°), and (0°, 0°) for 2D and 1D plots, respectively.

**Figure 15 sensors-24-06810-f015:**
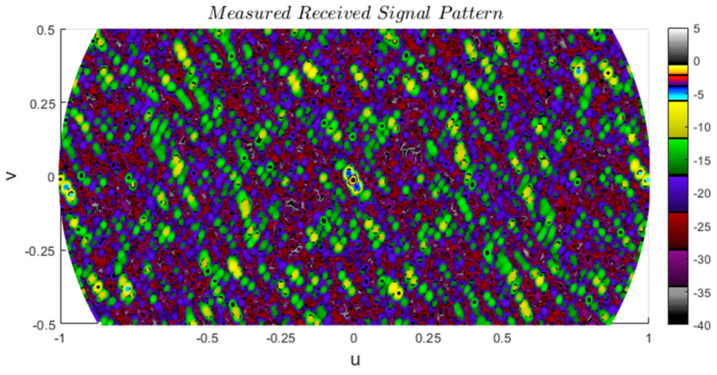
Measured received signal pattern for Ankara–2A with a single target at broadside.

**Figure 16 sensors-24-06810-f016:**
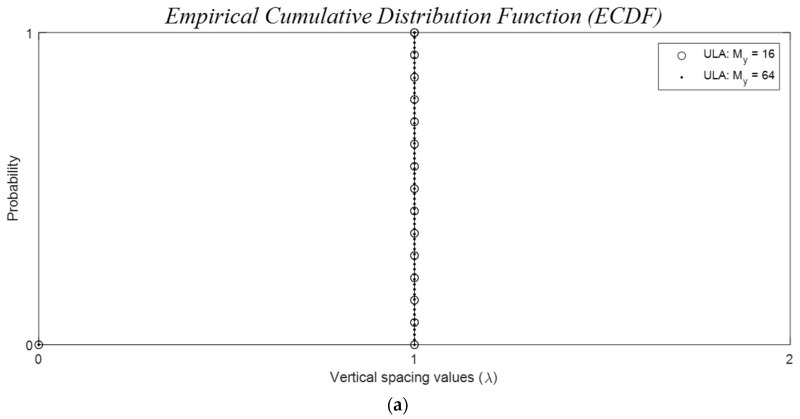
The empirical cumulative distribution functions (ECDF) for the horizontal inter-element spacings of virtual arrays with physical antenna elements of width λ. (**a**) Horizontal ULA with the number of elements of 16 and 64, and (**b**) the grid-based sparse Ankara arrays with versions A and B.

**Table 1 sensors-24-06810-t001:** Common notations used throughout the paper.

$\alpha_{ap},\;\alpha_{bw,\phi },\;\alpha_{bw,\theta }$	Aperture Loss and Beamwidth Spreading Factors
b,B $\mathrm{and}\;b_{s},B_{s}$	Beamforming vectors and matrices
$\beta \left(t\right)$	Reference phase at the origin
$C,C_{s}$	Calibration matrices; standard and for the sparse array formulation
$d,\;d_{\lambda },\Delta d$	Inter-element spacing, its value normalized to one wavelength and incremental spacing value
$d_{v}^{+}$ *,* $d_{v}^{-}$	The desirability functions for the LTB and STB cases
$f_{tx},f_{rx}$	Antenna elements gain factors
*FOV, uFOV*	Operational field of view (FOV) and the usable FOV
$\left(\phi_{t},\theta_{t}\right)$	The direction of the *t*’th target in spherical coordinates
$G,\;G_{s}$	Beamforming output
$\bar{k}$	Wavevector
$N_{tx},N_{rx},N_{vrx}$	Number of transmitter, receiver, and virtual receiver elements
p, P	Sparse array element order number and the total number of virtual elements
$q_{\phi },\;q_{\theta }$	Oversampling ratios for field domain $\left(u,\;v\right)$
$s_{s}$	Far-field received pattern
$r,\;r_{s}$	Steering vectors
$\bar{R},\;{\bar{R}}_{tx},{\bar{R}}_{rx}$	Displacement vectors for the field point, TX and RX
$\sigma_{c,t}$	Radar cross-section for the *t*’th target
t	Thinning/sparsity ratio
T	Number of targets
$\left(u,v\right)$	Directional cosine terms, $\left(\sin\phi \sin\theta ,\cos\theta \right)$ for the field domain
$\left(v_{min},v_{desired}\right)$	Expected observation range for the variable v
(x,y,z)	The field point based on the ISO/SAE coordinate system.
$\left(x^{\prime },y^{\prime },z^{\prime }\right)$	Source coordinates where the aperture is located on the $z^{\prime }=0$ surface.
$\left(y_{tx},z_{tx}\right),\left(y_{rx},z_{rx}\right)$	The transmitter and receiver coordinates on the aperture
$\left(Y_{max},Z_{max}\right)$	Antenna aperture dimensions along *y*- and *z*-axis.

**Table 2 sensors-24-06810-t002:** Inter-element spacings and the usable FOV values.

*Spacings* (λ)	0.5	0.5077	0.5321	0.5774	0.6527	0.7778	1	2	3	4	5	10	20
*uFOV* (deg)	180	160	140	120	100	80	60	28.96	19.19	14.36	11.48	5.73	2.87

**Table 3 sensors-24-06810-t003:** Aperture loss and beamwidth spreading factors.

	$\frac{y_{mp},\;z_{mp}}{\lambda }$	(a, b) Shared Fully Populated Aperture	(c, d) Vertical	(e, f) Diagonal	(g, h) Four Corners	(i, j) L Shaped Receivers	(k, l) Wrap-Around	(m, n) Improved Four Corners
*α_ap_*	(0, 0)	**1**	0.504	0.735	0.735	0.629	0.254	**1**
	(5, 5)	**1**	0.438	0.613	**0.681**	0.535	0.235	**0.670**
	(15, 10)	**1**	0.3398	**0.4897**	0.446	0.380	0.136	**0.490**
	(20, 15)	**1**	0.1746	**0.244**	0.142	0.199	–	**0.238**
*α_bw,ϕ_*	(0, 0)	**1**	**1**	**1.013**	**1**	**1**	2	**1**
	(5, 5)	**1**	**1**	**1.013**	**1**	1.067	2	**1**
	(15, 10)	**1**	**1**	**1.013**	**1**	1.231	2	**1**
	(20, 15)	**1**	**1**	**1.013**	**1**	1.455	–	**1**
*α_bw,θ_*	(0, 0)	**1**	2	1.008	**1**	1.333	2	**1**
	(5, 5)	**1**	2.308	1.212	**1**	1.395	2	**1**
	(15, 10)	**1**	3	1.519	**1**	1.500	2	**1**
	(20, 15)	**1**	6	3.078	**1**	1.714	–	**1**

**Table 4 sensors-24-06810-t004:** Inter-element mutual coupling between TX and RX elements for the sparse array Ankara.

	RX-1	RX-2	RX-3	RX-4	RX-5	RX-6	RX-7	RX-8	RX-9	RX-10	RX-11	RX-12	RX-13	RX-14	RX-15	RX-16
**TX-1**	−86.0	−77.5	−59.7	−55.6	−81.1	−80.6	−85.7	−72.5	−94.7	−91.4	−99.7	−101.4	−101.5	−109.3	−116.2	−110.9
**TX-2**	−41.5	−51.0	−50.1	−56.5	−67.7	−71.1	−60.7	−91.4	−63.8	−92.1	−70.8	−71.5	−97.7	−81.4	−91.4	−81.0
**TX-3**	−39.7	−41.1	−52.2	−79.1	−89.1	−75.5	−60.6	−77.9	−64.8	−91.6	−72.0	−69.0	−81.7	−77.0	−84.5	−78.8
**TX-4**	−58.3	−68.0	−57.2	−42.3	−57.2	−54.0	−84.9	−66.9	−104.2	−59.9	−85.2	−61.8	−67.0	−69.0	−72.4	−71.9
**TX-5**	−60.4	−78.3	−63.2	−67.3	−36.9	−63.3	−42.8	−43.0	−53.8	−66.3	−59.3	−52.5	−84.8	−65.6	−93.3	−64.7
**TX-6**	−78.9	−90.7	−77.2	−53.2	−40.5	−70.3	−35.3	−50.0	−34.8	−66.9	−46.2	−70.5	−53.3	−73.0	−80.9	−81.7
**TX-7**	−70.2	−81.2	−87.2	−82.8	−63.3	−78.0	−53.1	−54.9	−42.3	−55.2	−36.4	−36.7	−42.0	−47.3	−68.4	−50.0
**TX-8**	−88.7	−77.1	−85.3	−86.1	−54.3	−74.7	−47.9	−52.6	−54.7	−56.3	−49.5	−40.0	−41.0	−66.8	−70.7	−71.7
**TX-9**	−94.8	−97.0	−73.7	−66.4	−64.2	−51.0	−67.3	−46.0	−66.7	−40.2	−58.9	−50.9	−49.9	−76.8	−76.4	−91.2
**TX-10**	−101.1	−101.6	−80.0	−72.7	−70.3	−71.8	−71.8	−54.9	−76.0	−54.1	−66.1	−59.6	−55.3	−80.0	−81.2	−88.1
**TX-11**	−101.9	−99.9	−101.9	−92.4	−71.0	−62.0	−71.6	−84.7	−56.8	−57.6	−57.7	−52.9	−43.4	−51.5	−43.5	−48.5
**TX-12**	−94.0	−85.3	−105.7	−80.0	−80.8	−75.1	−72.1	−55.2	−85.8	−83.9	−52.0	−60.0	−47.7	−42.5	−31.5	−39.7

**Table 5 sensors-24-06810-t005:** Performance parameter values for example GBSAs.

	Ankara–1		Ankara–2	
	A	B	A	B
PSLR (dB)	11.23	8.64	11.10	9.14
BW-azimuth (deg)	0.55	0.53	0.4	0.5
BW-elevation (deg)	0.49	0.5	0.76	0.73
# of elements (TX, RX)	12, 16	12, 8	12, 16	12, 8
# of VRX’s (generated, unique)	192, 192	96, 96	192, 190	96, 96
Thinning ratio (%)	2.6	1.3	1.09	0.68
uFOV (deg)	180	60	180	180
Reference URA size	121 × 61		156 × 112	130 × 109
# reference URA elements	7381		17,472	14,170
Reference grid size (λ)	0.5, 1.0		0.5, 0.5	
Physical aperture size (λ)	32 × 35		41.5 × 34	

## Data Availability

Please see the link for the digital copies of figures, data and related code scripts, Figure 1, Figure 2, Figure 3, Figure 4, Figure 5, Figure 6, Figure 7, Figure 8, Figure 9, Figure 10, Figure 11, Figure 12, Figure 13, Figure 14, Figure 15 and Figure 16. Folder: Data-Sensors https://drive.google.com/drive/u/0/folders/1xSsMqpVWtbVMm7yAhre5kvWA2xMJ3bz2 (accessed on 14 October 2024).

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
