# Peer review of "Multi-Objective Design and Optimization of Hardware-Friendly Grid-Based Sparse MIMO Arrays"

_sensors, 2024, doi:10.3390/s24216810_

Round 1
Reviewer 1 Report
Comments and Suggestions for Authors
The paper proposes a comprehensive design framework for optimizing hardware-friendly, grid-based sparse MIMO arrays to enhance multi-target detection and reduce inter-element mutual coupling. The novelty lies in using grid-based sparse array configurations and an adaptive desirability function, along with a machine learning initialization approach, to enhance angular beamforming metrics and rapid convergence. In my opinion the work fits within the scope of Sensors and should be published after major revisions. In the following are several recommendations and clarifications that should be addressed before publication.
1. The introduction is structured clearly. However, it could further emphasize the study's innovative aspects and main contributions. When describing the current research landscape and technological background, it would be beneficial to clearly highlight the problems this research addresses and its distinctiveness.
2. It is recommended to provide a detailed description of the theoretical basis and implementation details of the machine learning initialization method, including model selection, training data, and hyperparameter optimization. This is one of the paper's key innovations, and a thorough explanation would enhance the method's reproducibility and academic value, enabling other researchers to better understand and apply it.
3. It is suggested to include a discussion on practical applications in the conclusion section. By linking the research to real-world applications, readers can better grasp the practical significance and potential impact of this work, which is crucial for advancing the practical implementation of the technology. For example, can the presented methods be processed in-situ by a diffractive neural network? The following papers are for reference. (https://doi.org/doi:10.1126/sciadv.ado3937; https://doi.org/10.1038/s41467-024-51210-2)
Author Response
Thank you very much for taking the time to review this manuscript. Please find the detailed responses attached and the corresponding revisions/corrections highlighted/in track changes in the re-submitted files. Detailed descriptions of the revisions are outlined in the attached Excel file (SENSORS-Reviewer-Requirements.xlsx), which highlights specific changes made throughout the manuscript.

Reviewer 2 Report
Comments and Suggestions for Authors
This article proposes a comprehensive design framework for optimizing sparse MIMO (multiple input multiple output) arrays to enhance multi-target detection, and introduce a new machine learning initialization method with special emphasis on grid-based antenna element configuration. The proposed method solves the key problems of multi-objective optimization, considering higher PSLR, smaller bandwidth, grating lobe challenges caused by utilizing larger antenna elements, and minimizing mutual coupling. In my view, this work falls within the scope of Sensors and should be published after making some modifications. A few suggestions and clarifications that need to be addressed before publication are presented below:
1. The description of the structure of the article needs to be more concise, which takes up too much space at present.
2. The font of the flowchart box in picture 1 is not agreed. Please make the font bold.
3. The serial number fonts in picture 3 are different. Please use bold font.
4. The font on the three pictures in picture 11 is too small, especially in picture (C), the font is not clear, please modify it.
5. In the last part of the conclusion, please appropriately add some future prospects and aspects that can be applied in the future.
Author Response
Thank you very much for taking the time to review this manuscript. Please find the detailed responses below and the corresponding revisions and corrections highlighted in track changes in the re-submitted files. The manuscript has been revised based on the constructive feedback from Reviewer 2. Key changes have been made to clarify the structure, figures, and prospects in the conclusion. In response to Reviewer’s comments, the structure of the article has been revised for conciseness. Unnecessary details have been minimized to make the paper more streamlined and focused. Further, based on the reviewer's suggestions, adjustments were made to the fonts in figures to ensure readability. For example, the font in the flowchart box in Figure 1 was made bold, and the serial number fonts in Figure 3 were adjusted for consistency. Additionally, the small, unclear fonts in Figure 11 (particularly subfigure (C)) were enlarged for better clarity.
Detailed descriptions of the revisions are outlined in the attached Excel file (SENSORS-Reviewer-Requirements.xlsx), which highlights specific changes made throughout the manuscript.

Reviewer 3 Report
Comments and Suggestions for Authors
The design and optimization of MIMO arrays is a very important issue. For enhancing enhance BW, PSLR, and grating lobe-free, this paper examines multi-objective design and optimization of hardware grid-based sparse MIMO arrays. So,this is interesting. However, the paper needs to be improved in the following areas:
1、Enhancing hardware design and optimization.
2、Analyzing the effect of standard sampling and under sampling on performance.
3、Analyzing the complexity of optimization algorithms.
4、Adjustment of the structure of Line 88 to Line 109.
5、English must be improved. For example, 'when the physical size of elements and so the spacings exceed this value' in Line 813.
Comments on the Quality of English Language
English must be improved. For example, 'when the physical size of elements and so the spacings exceed this value' in Line 813.
Author Response
Thank you very much for taking the time to review this manuscript. Please find the detailed responses below and the corresponding revisions/corrections highlighted/in track changes in the re-submitted files. Detailed descriptions of the revisions are outlined in the attached Excel file (SENSORS-Reviewer-Requirements.xlsx), which highlights specific changes made throughout the manuscript.
|

Round 2
Reviewer 3 Report
Comments and Suggestions for Authors
No
Comments on the Quality of English LanguageNo